# SLAM-family receptors promote resolution of ILC2-mediated inflammation

Yuande Wang [1,2], Yuhe Quan[2], Junming He[2], Shasha Chen[1,3,4,5] ✉ & Zhongjun Dong [1,2,3,4,5] ✉

Type 2 innate lymphoid cells (ILC2) initiate early allergic inflammation in the lung, but the factors that promote subsequent resolution of type 2 inflammation and prevent prolonged ILC2 activation are not fully known. Here we show that SLAM-family receptors (SFR) play essential roles in this process. We demonstrate dynamic expression of several SFRs on ILC2s during papain-induced type 2 immunity in mice. SFR deficiency exacerbates ILC2-driven eosinophil infiltration in the lung, and results in a significant increase in IL-13 production by ILC2s exclusively in mediastinal lymph nodes (MLN), leading to increased dendritic cell (DC) and TH2 cell numbers. In MLNs, we observe more frequent interaction between ILC2s and bystander T cells, with T cell-expressed SFRs (especially SLAMF3 and SLAMF5) acting as self-ligands to suppress IL-13 production by ILC2s. Mechanistically, homotypic engagement of SFRs at the interface between ILC2s and T cells delivers inhibitory signaling primarily mediated by SHIP-1. This prevents activation of NF-κB, driven by IL-7 and IL-33, two major drivers of ILC2-mediated type 2 immunity. Thus, our study shows that an ILC2-DC-TH2 regulatory axis may promote the resolution of pulmonary type 2 immune responses, and highlights SLAMF3/SLAMF5 as potential therapeutic targets for ameliorating type 2 immunity.

The lung inflammations mediated by type 2 immunity is primarily carried out by T helper 2 (TH2) cells and innate lymphoid cells (ILC2), which secrete specific cytokines, such as IL-5 and IL-13[1–3]. The receptors expressed on ILC2s coordinate crucial regulatory functions by perceiving microenvironmental stimuli[4], including cytokines (e.g., IL-33, IL-25, IL-7, interferon), metabolites (e.g., leukotrienes, prostaglandins), neural signals (e.g., neuropeptides, catecholamines, acetylcholine), and cell-to-cell interactions. Importantly, activated ILC2s exhibit high amoeboid-like motility in vivo[5], indicating frequent intercellular contacts. Indeed, direct receptor-mediated cell-to-cell communication between ILC2s and adjacent cellular populations plays a critical role in regulating ILC2 function. For instance, various receptors expressed on ILC2s such as ICOS, PD-1, KLRG1, LFA-1 and NKp30 can elicit regulatory

effects on ILC2 function through their respective ligands derived from intercellular interactions[6–11]. Regarding the interaction between ILC2s and T cells, MHC II, CD80, CD86 expression on ILC2s facilitates the activation of CD4+ T cells while OX40L expression promotes TH2 responses[12–14]. Moreover, expression of multiple ligands such as CD1a, PD-L1, ICOSL and GITRL on ILC2s also plays critical roles in modulating T cell responses[15–17]. Further investigation is needed to determine if T cells are involved in receptor-mediated crosstalk with ILC2s.

Upon exposure to stimuli such as papain or IL-33, pulmonary ILC2s undergo rapid activation, triggering the onset of initial pulmonary eosinophilia. As the inflammatory process advances, dendritic cells (DC) that capture antigens relocate from the lungs to the mediastinal lymph nodes (MLN), where they prime naïve T cells, thereby

[1]Department of Allergy, the First Affiliated Hospital of Anhui Medical University and Institute of Clinical Immunology, Anhui Medical University, Hefei 230032, China. [2]State Key Laboratory of Membrane Biology, School of Medicine and Institute for Immunology, Tsinghua University, Beijing 100084, China. [3]Innovative Institute of Tumor Immunity and Medicine (ITIM), Hefei 230032, China. [4]Anhui Province Key Laboratory of Tumor Immune Microenvironment and Immunotherapy, Hefei 230032, China. [5]Inflammation and Immune Mediated Diseases Laboratory of Anhui Province, Anhui Medical University, Hefei 230032, China. ✉ e-mail: chenshasha.26@163.com; dongzj@mail.tsinghua.edu.cn

amplifying subsequent T helper 2 (TH2) responses within the lung. Consequently, it is paramount to meticulously consider the immunological processes occurring within the MLNs, given their pivotal role in modulating subsequent lung inflammation. In line with findings in lungs where IL-13 secretion by ILC2s facilitates TH2 responses mediated by DCs[18,19], previous studies have also highlighted a substantial population of IL-13-secreting ILC2s in MLNs[19–23], suggesting a unique mechanism involving an ILC2-IL-13-DC axis specifically within these lymph nodes. Nevertheless, there is still limited understanding regarding the precise biological function of MLN ILC2s. A key challenge lies in selectively depleting these cells without affecting their lung counterparts as both MLN and lung ILC2s share similar expression profiles. Importantly, adaptive immune cells are more abundant in MLNs than lungs, implying a higher likelihood of cell-to-cell communication between ILC2s and T cells within these lymph nodes.

The signaling lymphocytic activation molecule (SLAM) family receptors (SFR), comprising seven members known as SLAMF1 to SLAMF7, recognize self-ligands, except for SLAMF4 which specifically binds to SLAMF2[24]. This family of receptors exhibits distinct expression patterns on the surfaces of various hematopoietic cells, including adaptive immune cells and ILC2s. SFRs can utilize the immune tyrosine switch motif (ITSM) to recruit SAP proteins or phosphatases such as SHP-1, SHP-2, and SHIP-1 in order to mediate activation or inhibition signals respectively. The interaction between SFRs and phosphatases is hindered by SAP binding. In other words, when SAP is absent from immune cells, SFRs recruit these phosphatases which impede immune cell functions[25]. SFRs play crucial roles in facilitating intercellular communications such as NK cell conjugation with target cells[26–28], T cell interaction with B cells in the germinal center[29], and double-positive thymocyte selection during NK-T cell development[30–32]. Previous investigations have demonstrated the essentiality of SAP in TH2 cells for TCR-induced GATA-3 expression and TH2 differentiation[33–35]. However, there is limited research on whether SFRs regulate the intercellular interaction between ILC2s and their bystander cells.

Here we show that deficiency of SFRs exacerbates ILC2-driven airway inflammation. Interestingly, SFRs specifically inhibit IL-13 production by ILC2s in the MLN but not in the lungs. The reduction of IL-13 production from ILC2s within the MLN mediated by SFRs leads to a decrease in DC numbers, resulting in a decline of DC-driven TH2 response. Our investigations also unveil a higher frequency of interactions between ILC2s and T cells in the MLN, and specific deletion of SFRs in T cells results in an increased expression of IL-13 by ILC2s within the MLN. Mechanistically, certain members of SFRs, particularly SLAMF3 and SLAMF5, suppress the downstream NF-κB pathway of IL-7/IL-33 signaling through SHIP-1 phosphatase. Therefore, within the MLN, SFR-mediated suppression of IL-13 production by ILC2s serves as a critical checkpoint for orchestrating inhibition of TH2 responses and ultimately contributes to resolution of airway inflammation.

## Results

### SFRs facilitate the resolution of type 2 airway inflammation
We conducted an analysis of scRNA-seq data from the GEO database (GSE102299 and GSE131996) to investigate the expression profile of SFRs on ILC2s. Our findings revealed that SLAMF2, SLAMF3, and SLAMF5 were consistently expressed on ILC2s, irrespective of their resting state or exposure to inflammatory stimuli such as house dust mite (HDM), IL-25, IL-33, Nippostrongylus brasiliensis (N.B.), or neuropeptide neuromedin U (NMU) (Supplementary Fig. 1a, b). Additionally, flow cytometry analysis confirmed the robust expression of SLAMF2, SLAMF3, and SLAMF5 in both lung and MLN ILC2s before or after intranasal administration of papain (Fig. 1a, b and Supplementary Fig. 1c, d), thereby suggesting a potential regulatory role for these receptors in ILC2 function.

Papain-induced eosinophilia is dependent on ILC2s[36]. To evaluate the role of SFRs in ILC2-driven inflammation, we intranasally

administered papain to both wild-type (WT) and SFR-knockout (SFR$^{-/-}$) mice (Supplementary Fig. 1c). Lung pathological staining on day 3 after papain administration showed comparable inflammatory cellular infiltration between WT and SFR$^{-/-}$ mice (Fig. 1c). However, we observed an exacerbation of inflammatory cellular infiltration in SFR$^{-/-}$ mice on days 6 and 9 after papain administration (Fig. 1c), along with an increased total cell counts in bronchoalveolar lavage fluid (BALF) on day 9 after papain administration in SFR$^{-/-}$ mice (Fig. 1d). Furthermore, lung immunofluorescent images and flow cytometry revealed an increase in eosinophilia in SFR$^{-/-}$ mice (Fig. 1e, f and Supplementary Fig. 1e, f). Our findings suggest that the absence of SFRs leads to a worsening of papain-induced inflammation.

Given that papain can induce the release of IL-33 to activate ILC2s[37], we investigated whether SFRs could inhibit eosinophilia induced by IL-33. Our findings revealed that a deficiency in SFRs resulted in an elevation of both the total number of cells in BALF and the count of eosinophils after administration of IL-33 (Fig. 1g, h). These data suggest that SFRs may promote resolving both papain- and IL-33-induced eosinophilia.

To enhance the physiological significance, an allergen model induced by Alternaria alternata (A. alternata) was conducted. We observed an increased number of total BALF cells and eosinophils in SFR-deficient mice (Fig. 1i), indicating the role of SFRs in mitigating eosinophilia. Additionally, a similar trend was observed in the sensitization/challenge model induced by papain or A. alternata (Supplementary Fig. 1g–i), further highlighting the essential involvement of SFRs in the modulation of eosinophilia.

### SFRs inhibit the production of ILC2 cytokines in the MLN
Next, we investigated whether the negative regulation of papain- or IL-33-induced eosinophilia mediated by SFRs is dependent on ILC2s. We examined the presence of ILC2s in both the lungs and MLN. Compared to WT mice, SFR$^{-/-}$ mice showed an increased number of ILC2s in the MLN on day 6 after papain or IL-33 administration, but no significant difference was observed in the lungs (Fig. 2a, b and Supplementary Fig. 2a). Furthermore, in SFR-deficient mice, there was also an elevated production of IL-5 and IL-13 by ILC2s specifically in the MLN on day 6, while no changes were detected in the lungs (Fig. 2c–f and Supplementary Fig. 2b). Consequently, it can be inferred that SFRs may exert a negative influence on both the quantity and cytokine production of ILC2s, particularly within the MLN.

The development of ILCs relies on Nfil3[38,39]. To confirm the dependence of SFR-regulated eosinophilia resolution on ILC2s, we generated Nfil3 SFR double knockout (SFR$^{-/-}$Nfil3$^{-/-}$) mice to deplete ILCs in SFR$^{-/-}$ mice (Supplementary Fig. 2c). In the absence of ILCs, papain or IL-33 did not induce increased eosinophilia in SFR$^{-/-}$ mice (Fig. 2g, h).

To exclude the possibility of involvement of ILC1s and ILC3s in SFR deficiency-induced eosinophilia, SR3335[40,41], an inhibitor for RORα, which is essential for the development, homeostasis, and effector function of ILC2s but not ILC1s and ILC3s, was used to deplete ILC2s in SFR$^{-/-}$ mice. The treatment significantly reduced the number of ILC2s (Supplementary Fig. 2d) and attenuated the increased eosinophilia (Fig. 2i, j) caused by SFR deficiency. Furthermore, adoptive transfer experiments were performed using WT and SFR-deficient ILC2s into Nfil3$^{-/-}$ recipients. The results demonstrated that the transfer of SFR-deficient ILC2s led to exacerbated eosinophilia compared to the transfer of WT ILC2s (Supplementary Fig. 2e). Therefore, over-activation of ILC2s contributes to unrestrained type 2 inflammation in SFR-deficient mice.

### FTY720 treatment prevents ILC2-mediated inflammation
FTY720, an FDA-approved immunomodulator, effectively reduces the circulation of lymphocytes by inhibiting S1P1-dependent lymphocyte egress from secondary lymphoid tissues such as MLN. In order to further validate the hypothesis that MLN is where SFRs inhibit ILC2 expansion and activation, and in turn hamper pulmonary inflammation

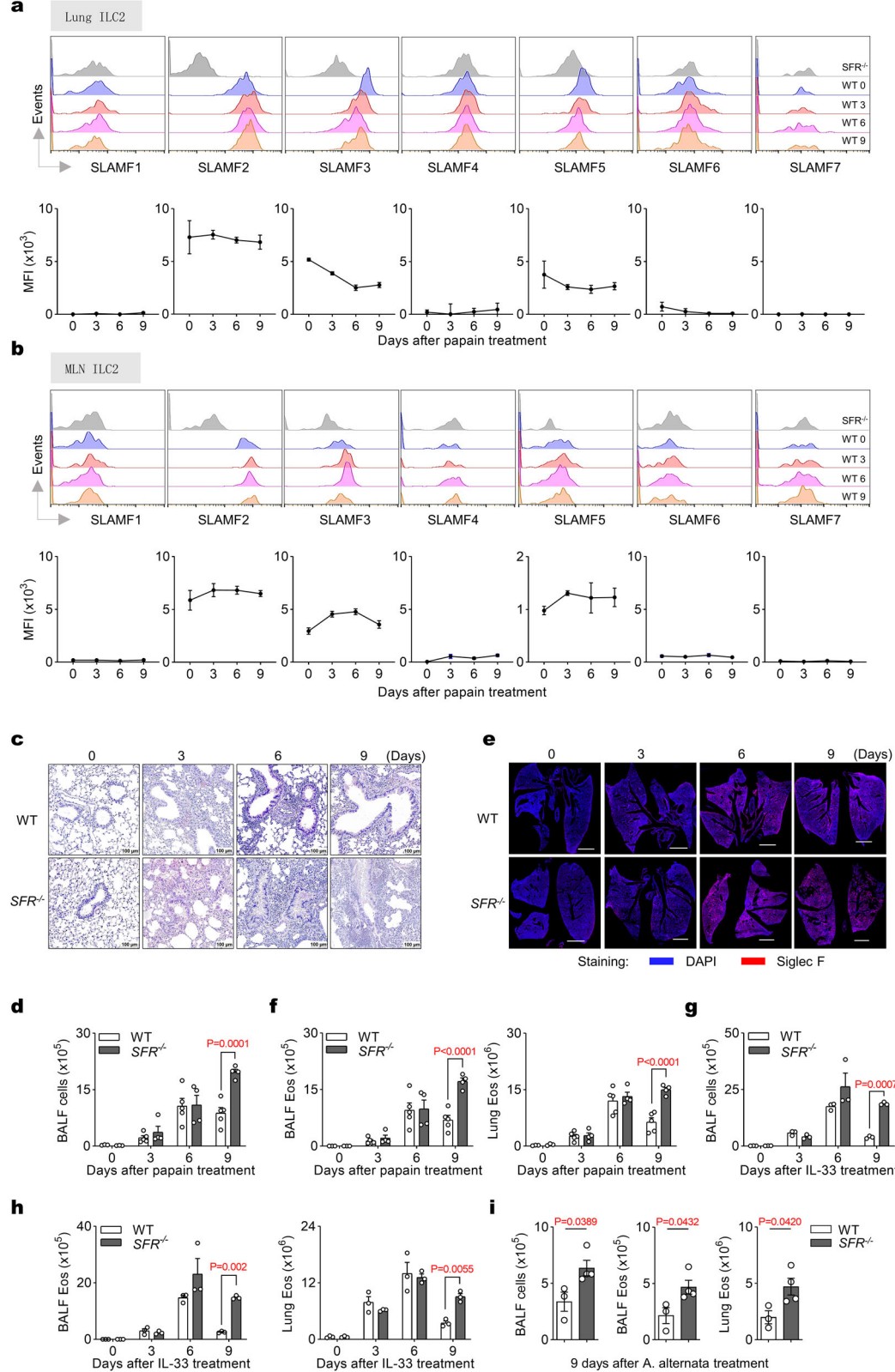

indirectly, we investigated whether interfering with MLN lymphocyte migration to the lungs could alleviate the severe pulmonary inflammation caused by SFR deficiency. Treatment with FTY720 resulted in a significant reduction in the number of CD4[+] T cells (Fig. 3a and Supplementary Fig. 3a) and eosinophils (Fig. 3b and Supplementary Fig. 3b) in the lungs of papain or IL-33-treated *SFR*[-/-] mice. Therefore, it can be concluded that lymphocyte egress plays a critical role in ILC2-

dependent pulmonary inflammation caused by SFR deficiency, suggesting potential involvement of MLN.

## ILC2 activation in MLN promotes TH2 cell-mediated lung inflammation

As the lymph nodes play a crucial role in T cell priming, we hypothesized that SFR-mediated suppression of ILC2s could indirectly hinder

**Fig. 1 | SFRs promote the resolution of airway inflammation. a, b** Mice were subjected to intranasal (i.n.) administration of papain from day 0 to day 2, as illustrated in Supplementary Fig. 1c. Representative overlaid histograms and the normalized expression levels of the SFR members on ILC2s in the lungs (**a**) and MLN (**b**) were performed using flow cytometry on the designated days. $n$(SLAMF1, SLAMF2, SLAMF4, SLAMF5, SLAMF7) = 3 mice per group; $n$(SLAMF3) = 3, 5, 5, 5 and $n$(SLAMF6) = 4, 3, 3, 3 mice in groups shown from left to right (**a**); $n$(SLAMF3, SLAMF5, SLAMF6, SLAMF7) = 3 mice per group; $n$(SLAMF2) = 6, 8, 8, 8 and $n$(SLAMF1 and SLAMF4) = 3, 5, 5, 5 mice in groups shown from left to right (**b**). **c** Representative H&E staining of the lung tissue from WT and $SFR^{-/-}$ mice on the indicated days after papain treatment (i.n., day 0, 1 and 2). **d** BALF cell numbers from WT and $SFR^{-/-}$ mice on the indicated days after papain treatment (i.n., day 0, 1 and 2). $n$(WT) = 3, 5, 5, 5 and $n$($SFR^{-/-}$) = 3, 4, 4, 4 mice in groups shown from left to right. **e** Representative panoramic view illustrating the lungs immunostaining of eosinophils from WT and $SFR^{-/-}$ mice on the indicated days after papain treatment (i.n., day 0, 1 and 2). Scale bar = 2000 µm. **f** The number of eosinophils in BALF and lungs from WT and $SFR^{-/-}$ mice on the indicated days after papain treatment (i.n., day 0, 1 and 2). $n$(WT) = 3, 5, 5, 5 and $n$($SFR^{-/-}$) = 3, 4, 4, 4 mice in groups shown from left to right. The number of BALF cells (**g**) and eosinophils (**h**) from mice on the indicated days after IL-33 treatment (i.n., day 0, 1 and 2). $n$ = 3 mice per group. **i** The number of BALF cells and eosinophils from WT and $SFR^{-/-}$ mice on day 9 after A. alternata treatment (i.n., day 0, 1 and 2). $n$(WT) = 3 and $n$($SFR^{-/-}$) = 4 mice. The data represent at two independent experiments with similar results. All data are represented as means ± SEM, and statistical analysis was performed using a two-way ANOVA (**d**, **f**–**h**) or two-tailed Student's $t$ test (**i**). Please refer to Supplementary Fig. 1 for further details.

the activation of CD4+ T cells within the MLN, subsequently leading to downregulation of CD4+ T cell-mediated pulmonary inflammation. To further investigate whether the enhanced pulmonary inflammation in $SFR^{-/-}$ mice is dependent on CD4+ T cells, we utilized SFR Rag1 double knockout ($SFR^{-/-}Rag1^{-/-}$) mice lacking both T- and B-cells. The absence of T- and B-cells in $SFR^{-/-}$ mice resulted in a significant reduction in eosinophil counts (Fig. 3c and Supplementary Fig. 3c). Furthermore, depletion of CD4+ T cells using a specific antibody targeting CD4 (clone GK1.5) also confirmed the involvement of CD4+ T cells in ILC2-dependent pulmonary inflammation caused by SFR deficiency (Fig. 3d and Supplementary Fig. 3d).

To further validate this conclusion, we also observed an increased presence of GATA-3-positive TH2 cells during the late stage of papain, IL-33 or A. alternata administration in $SFR^{-/-}$ mice (Fig. 3e and Supplementary Fig. 3e, f). The rise in eosinophil counts in $SFR^{-/-}$ mice strongly correlated with the number of TH2 cells (Fig. 3f and Supplementary Fig. 3g). CD69, an early activation marker for lymphocytes, exhibited elevated levels in $SFR^{-/-}$ CD4+ T cells (Fig. 3g and Supplementary Fig. 3h, i). Additionally, similar phenotypes were observed in the sensitization/challenge model induced by papain or A. alternata, including TH2 cells and CD69+ CD4+ T cells (Supplementary Fig. 3j, k). In order to measure antigen-specific CD4+ T cells, peptides 2W1S or OVA323 were utilized. Notably, there was a significant increase in tetramer-labeled CD4+ T cells within the lungs of $SFR^{-/-}$ mice following papain or IL-33 injection (Fig. 3h, i and Supplementary Fig. 3l). Collectively, these findings suggest that TH2 cell responses contribute to uncontrolled lung inflammation resulting from SFR deficiency.

The downstream adaptor SAP, which is recruited by SFRs, has previously been documented as a facilitator of TH2 differentiation[33-35]. Our consistent data, not presented here, indicates that the signaling pathway of SFR-SAP promotes TH2 differentiation in an OVA-induced model highly dependent on TH2 cells. Our perplexing findings lead us to hypothesize that SFRs indirectly suppress the TH2 response by inhibiting ILC2s, especially in papain or IL-33-induced models where ILC2s play a predominant role. Four approaches were employed to support this hypothesis. Firstly, we observed that genetic deletion of Nfil3 significantly reduced the number of TH2 cells, CD69+ CD4+ T cells, and antigen-specific tetramer+ CD4+ T cells in $SFR^{-/-}$ mice after papain or IL-33 treatment (Fig. 3j and Supplementary Fig. 3m). Chemical inhibition and deletion of ILC2s replicated the effects seen with Nfil3 deficiency (Fig. 3k and Supplementary Fig. 3n). Additionally, after papain treatment, the transfer of SFR-deficient ILC2s into $Nfil3^{-/-}$ mice led to higher numbers of TH2 cells and CD69+ CD4+ T cells compared to the transfer of WT ILC2s (Supplementary Fig. 3o). Finally, we aimed to generate mice with conditional deletion of SFRs on ILCs using a mixed bone marrow chimera assay[12] (Supplementary Fig. 3p). The resulting chimeras with Nfil3-deficient plus SFR-deficient bone marrow had over 90% of ILC2s that were negative for SFR expression (Supplementary Fig. 3q). After administration of peptide 2W1S plus IL-33, there was a significant increase in the proportion of TH2 cells, CD69+ CD4+ T cells, and tetramer+ CD4+ T cells only in the mixed chimeras

exhibiting SFR deficiency in ILC2s (Supplementary Fig. 3r). This series of evidence strongly suggests that the overactivated TH2 response observed in papain or IL-33-treated mice lacking functional SFRs relies on the presence of ILC2s.

## The production of IL-13 by ILC2s is inhibited by SFRs in MLN, which attenuates TH2 responses

The subsequent issue to be addressed pertains to the mechanism through which ILC2s elicit TH2 cell activation in MLN. Initially, we excluded the involvement of MHC II and IL-4 in regulating CD4+ T cells by ILC2s in MLN (Supplementary Fig. 4a, b). Previous studies have established that ILC2-derived IL-13 in the lungs can indirectly drive the TH2 response via DCs[18,19]. Considering that SFR-deficient mice exhibited elevated IL-13 production by ILC2s specifically in the MLN (but not in the lungs), along with an enhanced CD4+ T-cell response in the lungs, we hypothesized that ILC2-derived IL-13 within the MLN also positively contributes to DC activation, thereby promoting CD4+ T cell response. Consistent with the phenotype of ILC2s (Fig. 2a–f), SFR deficiency led to an increase in DC population, particularly DC2s, within the MLN but not within the lungs (Fig. 4a–d and Supplementary Fig. 4c–e). To investigate this further, we employed a CD11c-DTR genetic model where administration of diphtheria toxin (DTx) induced depletion of DCs (Supplementary Fig. 4f). Treatment with papain or IL-33 did not significantly exacerbate eosinophilia in DC-depleted $SFR^{-/-}$ mice (Fig. 4e, f). Notably, deficiency of DCs significantly attenuated the elevation observed for TH2 cells, CD69+ CD4+ T cells, and antigen-specific tetramer+ CD4+ T cells in SFR-deficient mice (Fig. 4g, h). These findings underscore that DCs are indispensable for driving severe lung inflammation resulting from TH2 responses induced by SFR deficiency.

The expression of SFRs is widely observed on almost all hematopoietic cells. In order to provide a clear explanation for the intrinsic or extrinsic role of SFRs in dendritic cells (DC), an in vitro assay was conducted, wherein wild-type or SFR-deficient bone marrow-derived DCs (BMDC) were co-cultured with OT-II TCR transgenic CD4+ T cells. It was observed that SFR-deficient BMDCs exhibited a comparable ability to induce TH2 polarization in CD4+ T cells (Supplementary Fig. 4g).

We further exclude the intrinsic role of SFRs in DCs by generating BM chimeras[19] (Supplementary Fig. 4h). The resulting chimeras exhibited a lack of SFR expression in approximately 80% of DCs following DTx administration (Supplementary Fig. 4i). Following the stimulation of peptide 2W1S plus IL-33, the proportion of TH2 cells, CD69-positive CD4+ T cells, and tetramer+ CD4+ T cells was comparable between two chimeras with DCs being SFRs positive or negative (Supplementary Fig. 4j). These findings suggest that extrinsic regulation of DC accumulation by SFRs is evident in our current model.

To determine whether the observed increase of DCs in the MLN of $SFR^{-/-}$ mice is dependent on ILC2s, we administered intranasally papain or IL-33 to $SFR^{-/-}Nfil3^{-/-}$ mice and found that deficiency in ILCs abolished the increase in numbers of DCs as well as DC2s within the MLN of $SFR^{-/-}$ mice (Fig. 4i, j). Furthermore, following papain treatment, the

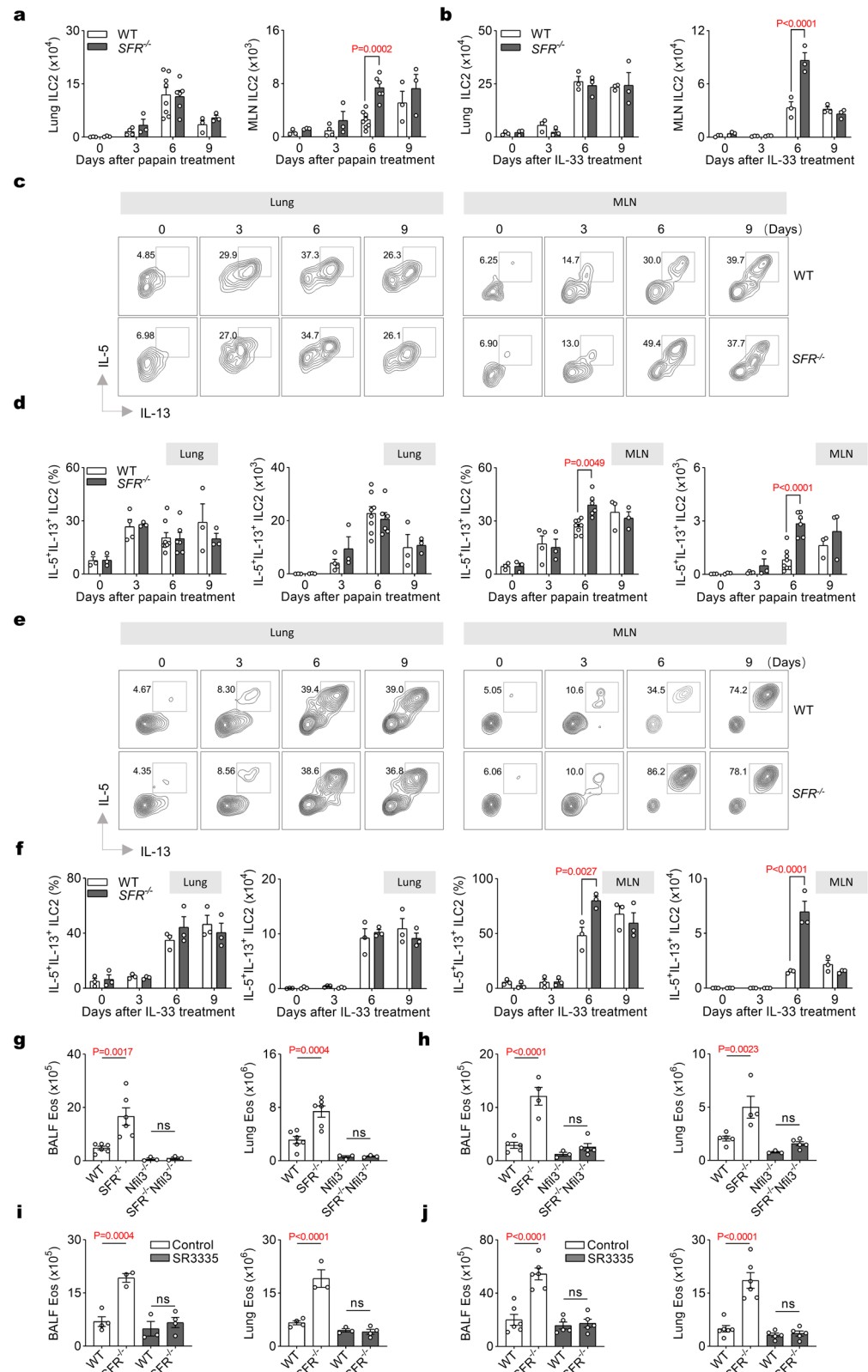

infusion of SFR-deficient ILC2s into *Nfil3⁻/⁻* mice resulted in an increased number of DCs compared to the transfer of WT ILC2s (Supplementary Fig. 4k). Thus, ILC2 plays a crucial role in the enhanced accumulation of MLN DCs in SFR-deficient mice.

To validate the significance of IL-13 in SFR-mediated resolution of pulmonary inflammation, IL-13-deficient mice were generated (Supplementary Fig. 4l, m). The results demonstrated that the

absence of IL-13 prevented SFR-deficient mice from increasing the number of DCs in MLNs on day 6 post-papain or IL-33 treatment, compared to WT controls (Fig. 4k, l). Similarly, in the absence of IL-13, SFR deficiency failed to exacerbate eosinophilia (Fig. 4m, n) as well as increase the number of TH2 cells, CD69-positive CD4⁺ T cells, and tetramer-positive CD4⁺ T cells (Fig. 4o, p). Collectively, these findings suggest that SFRs inhibit MLN DCs to promote CD4⁺ T cell

**Fig. 2 | SFRs inhibit ILC2 cytokine production in MLN.** Mice received daily intranasal injections of papain (**a**) or IL-33 (**b**) from day 0 to day 2. The number of ILC2s in lungs and MLN was quantified on the designated days. $n$ = 3, 3, 4, 3, 8, 6, 3, 3 mice in groups shown from left to right (**a**); $n$ = 3 mice per group (**b**). Representative flow cytometric profiles (**c**) and quantification (**d**) showing IL-5 and IL-13 expressing in gated lung and MLN ILC2s after 3 h of PMA plus ionomycin restimulation on the indicated days following papain treatment (i.n., day 0, 1 and 2). $n$ = 3, 3, 4, 3, 8, 6, 3, 3 mice were examined for groups shown from left to right. Representative flow cytometric profiles (**e**) and quantification (**f**) showing IL-5 and IL-13 expressing in gated lung and MLN ILC2s after 3 h of PMA plus ionomycin restimulation on the indicated days following IL-33 treatment (i.n., day 0, 1 and 2).

$n$ = 3 mice per group. Quantification of eosinophils in BALF and lungs on day 9 after mice were treated with papain (**g**) or IL-33 (**h**) on day 0, 1, 2. $n$ = 6, 6, 3, 3 mice (**g**) or $n$ = 5, 4, 3, 5 mice (**h**) were examined for groups shown from left to right. **i, j** WT and $SFR^{-/-}$ mice were intraperitoneally treated with either SR3335 or DMSO (control). The number of eosinophils in BALF and lungs was quantified on day 9 after papain (**i**) or IL-33 (**j**) administration (i.n., day 0, 1 and 2). $n$ = 4, 3, 3, 4 mice (**i**) or $n$ = 6, 6, 5, 5 mice (**j**) were examined for groups shown from left to right. The data represent at least three independent experiments with similar results. All data are represented as means ± SEM, and statistical analysis was performed using two-way (**a, b, d, f**) or one-way (**g–j**) ANOVA. ns, not significant. Please refer to Supplementary Fig. 2 for further details.

responses possibly by direct suppression of ILC2-mediated production of IL-13 in MLNs.

## ILC2 activation is inhibited by the bystander T cells in the MLN

Afterwards, we investigated the underlying mechanism by which SFRs inhibit ILC2 cytokine production in the MLN while having no effect in the lungs. Considering that most SFR members are self-ligands capable of activating or inhibiting SFR signals through cell-cell interaction, we examined the population of SFR-positive cells in the MLN. Flow cytometry analysis revealed that a majority of SFR-positive cells were adaptive immune cells (Fig. 5a and Supplementary Fig. 5a), and there was a significantly higher density of adaptive immune cells in the MLN compared to the lungs (Supplementary Fig. 5b, c). This suggests that adaptive immune cell-derived SFRs may have more opportunities to serve as ligands for activating SFR signaling on ILC2s in the MLN. To validate this hypothesis, we co-cultured WT ILC2s with either WT or $SFR^{-/-}$ T- or B-cells, respectively. The results demonstrated that WT ILC2s co-cultured with $SFR^{-/-}$ T or B cells exhibited significantly increased cytokine production compared to those co-cultured with WT T- or B-cells. Conversely, similar phenotypes could not be observed when mixing $SFR^{-/-}$ ILC2s with WT T or B cells (Fig. 5b, c and Supplementary Fig. 5d–f).

To assess the inhibitory effect of SFRs by adaptive immune cells on ILC2s in an in vivo model, we generated mixed chimeras by combining $Rag1^{-/-}$ BM with either WT or $SFR^{-/-}$ BM. This approach allowed us to obtain mice harboring WT or $SFR^{-/-}$ T- and B-cells (Supplementary Fig. 5g, h). Following administration of papain or IL-33, the absence of SFRs derived from adaptive immune cells led to increased production of IL-13 by ILC2s in the MLN (Fig. 5d, e). Despite the expression of SFRs on DCs, which were also located adjacent to ILC2s[42], our experiments using mixed bone marrow chimeras demonstrated that DCs lacking these receptors were unable to enhance IL-13 production by ILC2s (Supplementary Fig. 5i). These findings suggest that SFRs derived from T and/or B cells may play a role in inhibiting cytokine production by ILC2s.

To further investigate the inhibitory effects of T or B cell-derived SFRs on ILC2s, we employed confocal imaging to quantitatively visualize the localization of ILC2s in both the T zone and B zone of MLN. Interestingly, our findings revealed a significantly higher abundance of ILC2s in the MLN T zone compared to the B zone (Fig. 5f and Supplementary Fig. 5j). This distinction was further confirmed by using a specific NK1.1 antibody (clone PK136) to deplete NK cells in order to eliminate any potential interference from KLRG1+ NK cells during the identification of ILC2s (CD3-KLRG1+) through confocal imaging (Supplementary Fig. 5k, l). These results suggest that T-zone SFRs may play a crucial role in suppressing cytokine production by ILC2s.

To investigate the specific roles of bystander T cells in restricting ILC2 activation within the MLN niche, we sought to generate conditional SFR knockout mice ($SFR^{fl/fl}$) by inserting two LoxP sites into the slam loci flanks (approximately 400 kb) (Supplementary Fig. 5m). The resulting SFR floxed mice were then bred with *CD4-Cre* and/or *MB1-Cre* mice to selectively eliminate SFR expression in T- and/or B-cells.

Surpassing our expectations, flow cytometry analysis revealed that the efficiency of SFR deletion on T- and B-cells was nearly 80% in the MLN (Supplementary Fig. 5n, o). Following intranasal administration of IL-33, we observed a significant upregulation of ILC2 cytokine production in mice with T cells lacking SFRs, similar to that observed in germline SFR knockout mice (Fig. 5g). However, sole deletion of SFRs on B cells failed to upregulate ILC2 cytokine production and did not synergize with T-cell deletion of SFRs to enhance ILC2 activation (Fig. 5g). These genetic findings clearly underscore the essential inhibitory role played by T cell-expressed SFRs for ILC2 cytokine production within the MLN niche.

## ILC2 activation is inhibited by two major members of SFRs, namely SLAMF3 and SLAMF5

Our subsequent objective was to identify the key members of SFRs that primarily inhibit cytokine production by ILC2s. Initially, through a library of mice lacking each individual member of SFRs, we discovered that the loss of any single SLAM member did not significantly impact ILC2 cytokine production or eosinophilia (Fig. 6a and Supplementary Fig. 6a, b), indicating in vivo redundancy within the SFRs. To address this concern, we separately overexpressed each SFR member as ligands on SFR-null OP9-DL1 cell lines and co-cultured them with WT ILC2s. As a result, the ligands expressed on OP9-DL1 interacted with the corresponding SFRs expressed on ILC2s. We observed that only SLAMF3 or SLAMF5 among the seven members of SFRs was capable of preventing ILC2s from producing type 2 cytokines (Fig. 6b and Supplementary Fig. 6c). The immunofluorescence analysis revealed the polarization of SLAMF3 and SLAMF5 receptors at the contact surfaces between ILC2s and their bystander T cells (Fig. 6c and Supplementary Fig. 6d), indicating a potential role for these receptors in facilitating functional synapse formation between these immune cell types within the MLN niche. This prompted us to generate SLAMF3 and SLAMF5 double knockout ($SLAMF3/5^{-/-}$) mice to reveal any compensatory effects exerted by these two receptors (Supplementary Fig. 6e, f). Simultaneous loss of both receptors significantly enhanced ILC2 cytokine production in the MLN (Fig. 6d), as well as the numbers of eosinophils, TH2 cells, CD69+ CD4+ T cells, and antigen-specific CD4+ T cells in the lungs (Fig. 6e, f). In conclusion, our study suggests that two major members of SFRs, namely SLAMF3 and SLAMF5, are responsible for restricting ILC2 activation in the MLN.

## SLAMF3 and SLAMF5 inhibit ILC2 activation through phosphatase SHIP-1

SFRs play a crucial role in intercellular communications by recruiting SAP adaptor or SAP-independent phosphatases[24,25]. However, ILC2s lack the expression of SAP and its homologs, Ewing's sarcoma-associated transcript 2 (EAT-2) (Fig. 7a and Supplementary Fig. 7a). Indeed, the absence of adaptors, SAP and EAT-2, did not significantly enhance cytokine production by ILC2s (Fig. 7b), thereby supporting the conclusion that SFR-mediated suppression of ILC2 cytokine production is independent of SAP-related adaptors.

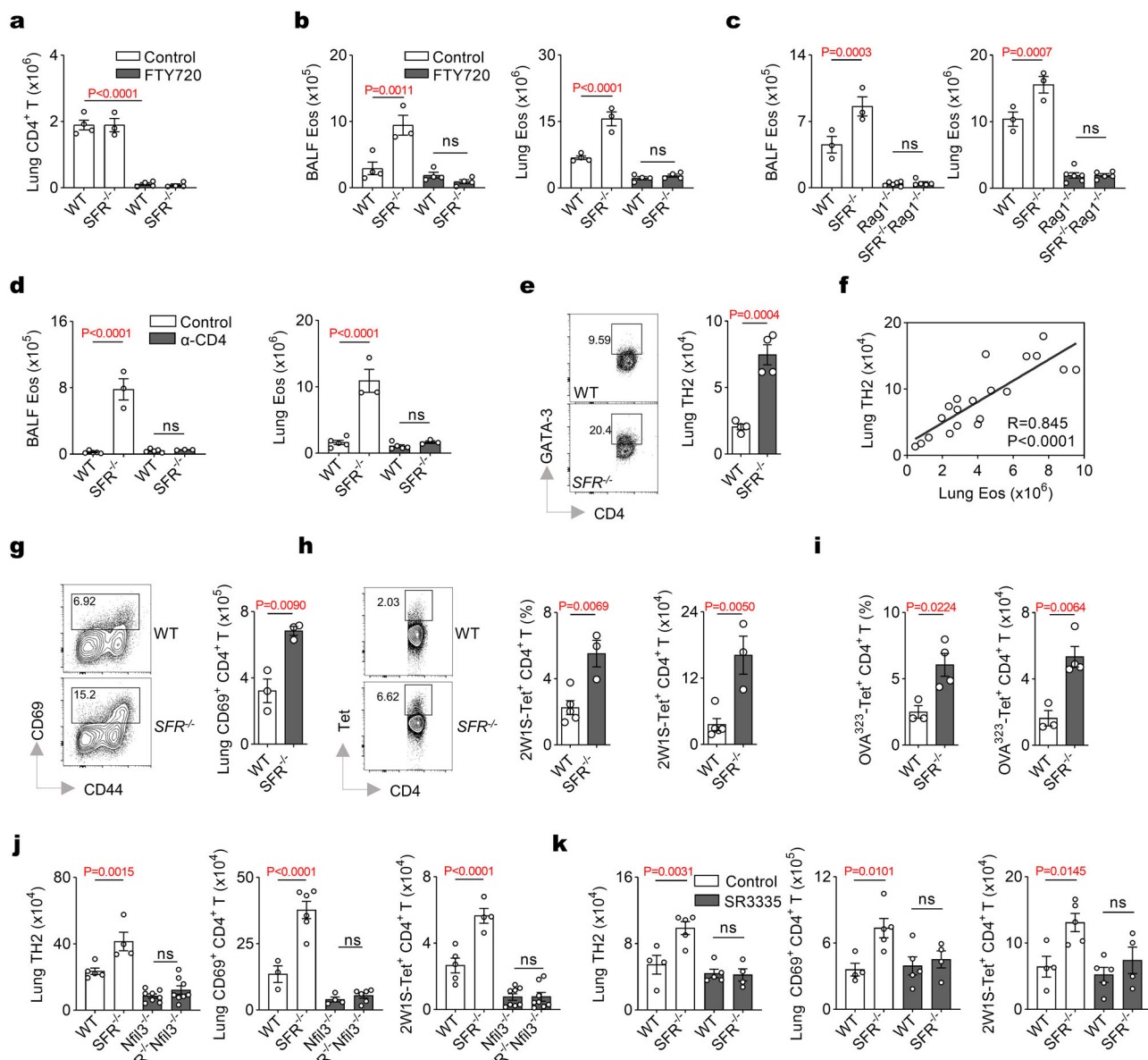

**Fig. 3 | The exacerbation of ILC2-mediated airway inflammation due to SFR-deficiency is dependent on TH2 response.** Detection of CD4+ T cells (**a**) and eosinophils (**b**) in DMSO (control) or FTY720-treated mice at day 9 following papain treatment (i.n., day 0 to 2). $n = 4, 3, 4, 4$ mice (left to right groups). **c** The number of eosinophils in mice at day 9 following papain treatment (i.n., day 0 to 2). $n = 3, 3, 6, 5$ mice (left to right groups). **d** The number of eosinophils on day 9 in the papain-induced (i.n., day 0 to 2) mice depleted of CD4+ T cells. α-CD4, anti-CD4 antibody; control, IgG1. $n = 5$ (WT) and 3 ($SFR^{-/-}$) mice per group. **e** The numbers of TH2 cells (CD3+CD4+GATA-3+) in the papain-induced (i.n., day 0 to 2) mice on day 9. $n = 4$ mice per group. **f** The correlation analysis between the numbers of TH2 cells and eosinophils in the lungs from WT and $SFR^{-/-}$ mice at day 9 following papain treatment (i.n., day 0 to 2). $n = 20$. **g** The number of CD69+CD4+ T cells in the papain-

induced (i.n., day 0 to 2) mice on day 9. $n = 3$ mice per group. The proportion and number of tetramer-traceable CD4+ T cells on day 9 after papain plus peptide 2W1S (**h**) or peptide OVA323 (**i**) treatment (i.n., day 0 to 2). $n$(WT) = 5 (**h**) or 3 (**i**) mice; $n$($SFR^{-/-}$) = 3 (**h**) or 4 (**i**) mice. **j** Detection of indicated cells in mice at day 9 following peptide 2W1S plus papain treatment (i.n., 0 to 2). $n = 5, 4, 8, 8, 3, 6, 4, 6, 5, 4, 8, 8$ mice (left to right groups). **k** Detection of indicated cells in SR3335-treated mice at day 9 following peptide 2W1S plus papain treatment (i.n., day 0 to 2). $n = 4, 5, 5, 4$ mice (left to right groups). The data represent at least three independent experiments with similar results. All data are represented as means ± SEM, and statistical analysis (±SEM) was conducted using one-way ANOVA (**a–d, j, k**), two-tailed Student's $t$ test (**e, g–i**), or linear regression (**f**). ns, not significant. Please refer to Supplementary Fig. 3 for further details.

When SAP is absent from immune cells, SFRs can recruit phosphatases such as SHP-1, SHP-2, and SHIP-1, which impede immune cell functions[24,25]. It is worth noting that while SHP-1, SHP-2, and SHIP-1 have been found to be expressed on ILC2s (Fig. 7c and Supplementary Fig. 7b), only the phosphorylation of SHIP-1 showed a significant decrease in $SLAMF3/5^{-/-}$ mice compared to WT controls (Fig. 7d–f). The cross-linking of SLAMF3 or SLAMF5 on ILC2s seemed to induce the phosphorylation of SHIP-1 (Fig. 7g and Supplementary Fig. 7c),

suggesting that a potential role of SHIP1 in mediating the inhibitory effects of SLAMF3 and SLAMF5 on ILC2s.

To strengthen the inhibitory function of SHIP-1 on ILC2 cytokine production, we utilized the Chimeric Immune Editing (CHIME) approach[43,44]. Our findings revealed a substantial enhancement in ILC2 cytokine production upon specific deletion of SHIP-1 using a single guide RNA (sgRNA, Fig. 7h). In summary, the involvement of SLAMF3 and SLAMF5 might be pivotal for triggering SHIP-1 activation and

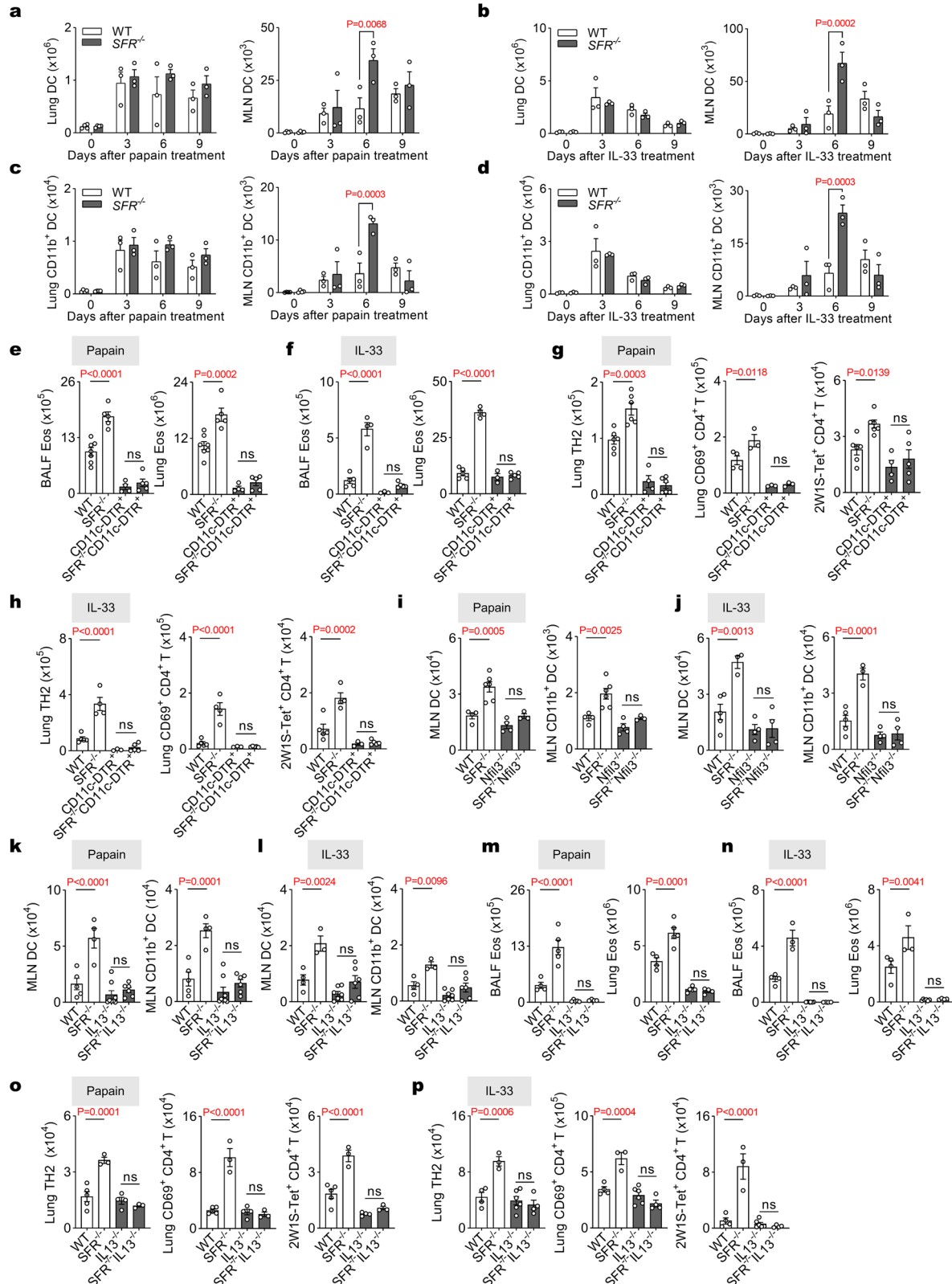

subsequent inhibition of ILC2 cytokine production within the MLN niche.

## SLAMF3 and SLAMF5 inhibit NF-κB pathway downstream IL-7/IL-33

Given the pivotal role of the NF-κB pathway in cytokine production by ILC2, as elucidated in previous studies[4,45,46], along with the demonstrated suppressive capability of SHIP-1 on the NF-κB pathway[47–49], our objective was to investigate whether SLAMF3 and SLAMF5 could potentially inhibit the NF-κB signal in ILC2s. We observed a reduction in expression of IκB, an inhibitor of NF-κB whose expression showed an inverse correlation with NF-κB activity[50], in MLN ILC2s lacking SLAMF3 and SLAMF5 (Fig. 7i). This finding was further supported by immunofluorescence studies revealing increased

**Fig. 4 | SFRs inhibit ILC2 IL-13 production, thereby attenuating the DC-TH2 axis in MLN.** The numbers of DCs (CD11c⁺MHC II⁺, **a**, **b**) or DC2s (CD11c⁺MHC II⁺CD11b⁺, **c**, **d**) in the lungs and MLN on the indicated days in mice following papain (**a**, **c**) or IL-33 (**b**, **d**) treatment for three consecutive days. $n = 4, 4, 3, 3, 3, 3, 3$ mice (left to right groups, **a**, **c**) or $n = 3$ mice per group (**b**, **d**). **e–h** Quantification of eosinophils (**e**, **f**), numbers of TH2 cells, CD69⁺CD4⁺ T cells, and tetramer⁺ CD4⁺ T cells (**g**, **h**) in DTx-treated mice at day 9 following peptide 2W1S plus papain (**e**, **g**) or IL-33 (**f**, **h**) treatment (i.n., day 0 to 2). $n = 7, 5, 4, 5$ (**e**), $n = 5, 4, 3, 5$ (**f**, **h**), $n = 6, 6, 4, 6, 4, 3, 3, 6, 6, 4, 5$ (**g**) mice in groups shown from left to right. **i**, **j** The numbers of DCs and DC2s in the MLN in mice on day 6 after papain (**i**) or IL-33 (**j**) treatment (i.n., day 0 to 2).

$n = 4, 6, 4, 3$ (**i**) and $n = 5, 3, 4, 4$ (**j**) mice (left to right groups). The numbers of DCs and DC2s in the MLN in mice on day 6 after papain (**k**) or IL-33 (**l**) treatment (i.n., day 0 to 2). $n = 5, 4, 8, 6$ (**k**) and $n = 4, 3, 7, 6$ (**l**) mice (left to right groups). Quantification of eosinophils (**m**, **n**), TH2 cells, CD69⁺CD4⁺ T cells and tetramer⁺ CD4⁺ T cells (**o**, **p**) on day 9 in mice treated with papain (**m**, **o**) or IL-33 (**n**, **p**) together with peptide 2W1S (i.n., day 0 to 2). $n = 4, 5, 5, 4$ (**m**), $n = 4, 3, 6, 4$ (**n**, **p**) and $n = 5, 3, 4, 3$ (**o**) mice (left to right groups). The data represent at least two independent experiments with similar results. All data are represented as means ± SEM, and statistical analysis was conducted using two-way (**a–d**) or one-way (**e–p**) ANOVA. ns not significant. Please refer to Supplementary Fig. 4 for further details.

phosphorylation of NF-κB in MLN ILC2s from *SLAMF3/5⁻/⁻* mice (Fig. 7j, k and Supplementary Fig. 7d). Additionally, stimulation with IL-7 or IL-33, two major drivers of ILC2 activation and expansion, resulted in significant phosphorylation of both IκB and NF-κB. However, crosslinking of SLAMF3 and SLAMF5 or antibody-stimulation effectively hindered this activation (Fig. 7l, m and Supplementary Fig. 7e–j). Therefore, it can be inferred that SLAMF3 and SLAMF5 inhibit the IL-7/IL-33 activated NF-κB pathway.

## Discussion

ILC2 is recognized for its ability to rapidly initiate type 2 immunity and enhance the downstream inflammatory responses by amplifying the TH2 response. It is worth noting that a reduction in inflammation often occurs during the later stages of ILC2-associated inflammation[18–20]. However, the precise inhibitory mechanism underlying the ILC2-mediated TH2 response and its contribution to resolving inflammation remain to be elucidated. In this study, we have identified the localization of ILC2s in the T zone of MLN. This enables the by-stander T cell-derived SLAMF3 and SLAMF5 to suppress IL-13 production by ILC2s, thereby impeding DC promotion towards CD4⁺ T cell responses and ultimately facilitating inflammation mitigation. Consequently, this investigation not only unravels SFRs responsible for negatively regulating ILC2 function but also sheds light on the mechanism of ILC2-mediated resolution of inflammation.

The immunological profiles of the lungs and MLN exhibit notable differences, particularly in terms of T cell abundance in the MLN, which provides a greater opportunity for interaction with ILC2s. Both ILC2s and T cells express self-ligands SLAMF3 and SLAMF5 on their surfaces, enhancing T cell-mediated inhibition of ILC2 cytokine production. Despite the presence of T cells in the lungs, the absence of SFRs fails to increase cytokine production by ILC2s. The low concentration of T cells in the lungs suggests that achieving a specific activation threshold for these cells to trigger inhibition of ILC2s is challenging. These findings highlight the impact of immune microenvironment diversity between the lungs and MLN on immune regulation of ILC2s.

Despite the substantial population of type 2 cytokine-producing ILC2s in MLN[19–23], elucidating their precise biological role remains challenging. This challenge arises from the shared expression profiles between ILC2 populations in the MLN and lungs, making it difficult to employ experimental techniques that specifically deplete MLN ILC2s without affecting their lung counterparts. In this study, we demonstrated that SFRs exert a specific inhibitory effect on ILC2 cytokine production within the MLN while leaving lung ILC2s unaffected. Consequently, utilizing a mouse model deficient in SFRs provides a valid approach to unraveling the biological functions of ILC2s within the MLN. By using this mouse model, we discovered that IL-13 derived from ILC2s enhances the number of DCs within the MLN and consequently facilitates the initiation of CD4⁺ T cell responses. However, further investigation is necessary to fully understand the underlying mechanisms involved in how IL-13 derived from ILC2s regulates DC accumulation and function in MLN. Thus, we identified the biological function of ILC2 in MLN, opening new avenues for investigating its role.

It is crucial to thoroughly comprehend the immune processes involving ILC2s and TH2 cells occurring within the lungs and MLNs at various time points to fully grasp the complexities of pulmonary type II immunity. Upon allergen inhalation, the rapid activation of ILC2s initiates early-stage lung eosinophilia, which typically precedes the involvement of TH2 cells. TH2 cell generation relies on antigen uptake by DCs in the lungs and subsequent presentation in the MLNs. Moreover, the activated TH2 cells need to migrate from the MLNs to the lungs to further exacerbate pulmonary inflammation. Thus, the immune processes taking place within the MLNs play a crucial role in the later stages of pulmonary inflammation, particularly regarding TH2 responses[51]. Indeed, our study observed comparable levels of inflammation during the early phase as there was no significant impact on ILC2 function within lungs of SFR-deficient mice. However, as inflammation progressed, we noted an increase in both ILC2 functions and DC counts in MLNs of SFR-deficient mice that corresponded with a similar phenomenon seen during subsequent lung inflammation. These findings highlight the importance of MLN immune processes in regulating lung inflammation. Nevertheless, further investigation is warranted to ascertain the origin of MLN ILC2s and the factors driving their localization to the T zone of MLNs.

In our investigation, despite our efforts to clarify the impact of SFRs on ILC2s in influencing the behaviors of TH2 cells and DCs, and our attempts to eliminate the inherent effects of SFRs on T cells and DCs, using the germline SFR knockout mouse model may introduce complexities. To address these limitations more accurately, it would be preferable to use mice with ILC2-specific deletion of SFRs. In pursuit of this, efforts were made to produce *SFRᶠˡ/ᶠˡ/IL-5-Cre* mice; however, due to the large size of floxed SFR loci (approximately 400 kb), deletion efficiencies were only observed with CD4-Cre and MB1-Cre among the various Cre mice used, including IL-5-Cre and CD11c-Cre. Nevertheless, ongoing efforts involve generating *SLAMF3ᶠˡ/ᶠˡ/SLAMF5ᶠˡ/ᶠˡ* mice for specific deletion of two identified major members of SFRs, followed by crossing them with *IL-5-Cre* or newly-generated *Nmur1-Cre* mice by two other groups[52,53].

Our research extends the role of ILC2s in TH2 immunity. The regulatory role of ILC2s in TH2 responses is of interest because it can increase the TH2 response and subsequent type 2 immunity. Expression of MHC II, CD80, and CD86 on ILC2s facilitates the activation of CD4⁺ T cells, while OX40L expression further promotes TH2 responses[12–14]. Additionally, the pulmonary ILC2s have been shown to contribute to DC-mediated TH2 responses by secreting IL-13[18], and it also recruit memory TH2 responses by inducing the release of the chemokine CCL17 by DCs[19]. This study proposes a mechanism that emphasizes the significance of IL-13 from ILC2s in MLN in initiating the TH2 responses.

In summary, this study reveals an unexpected mechanism whereby SFRs negatively regulate airway inflammation through modulating cellular interactions between ILC2s and their by-stander T cells within the T-zone of MLN (Supplementary Fig. 8). This study also highlights that the therapeutic enhancement of the inhibitory signaling mediated by SFR family receptors may be a potential new treatment approach for type II inflammation.

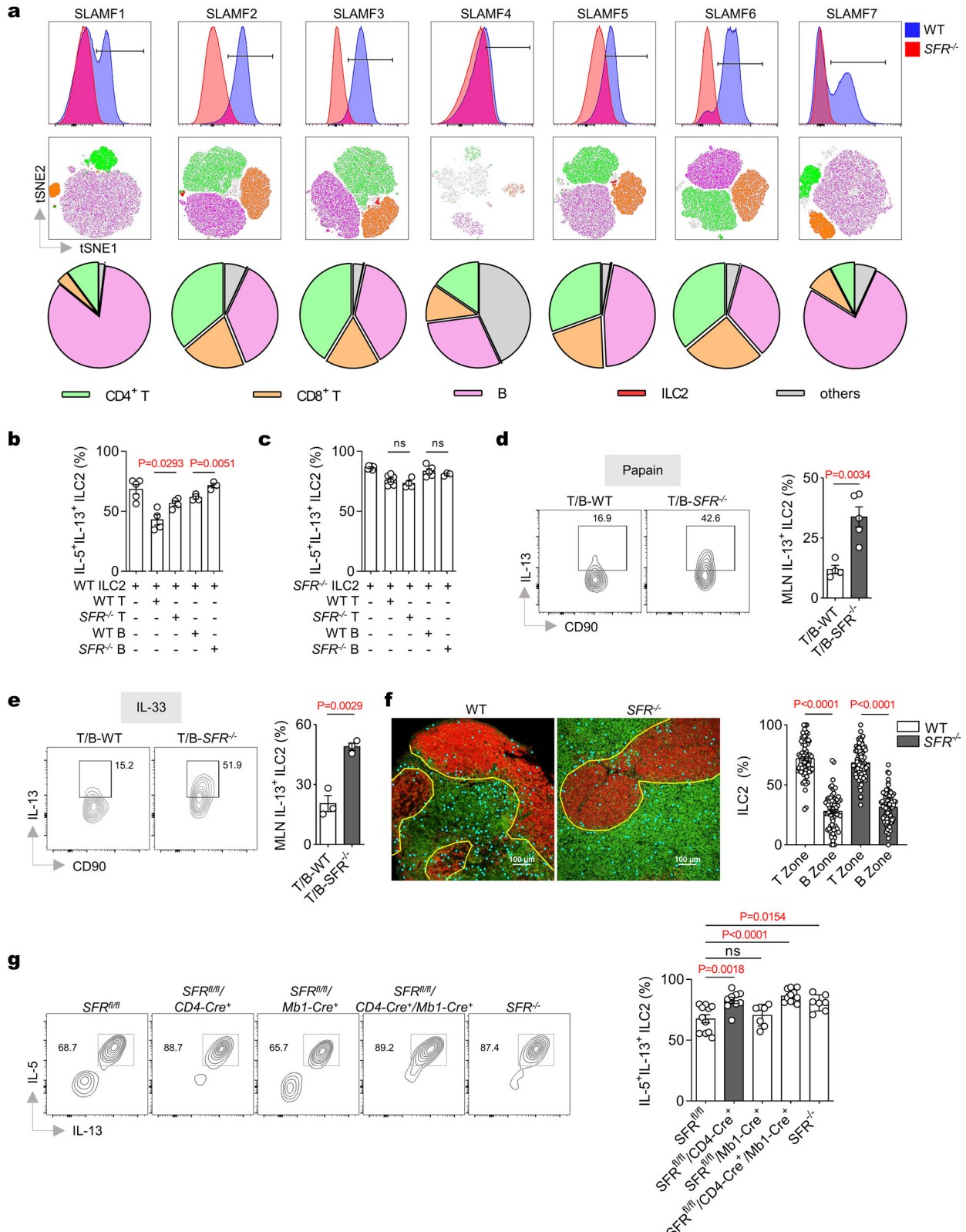

## Methods

### Mice
Mice lacking SLAM family members ($SFR^{-/-}$), SLAMF1-deficient ($SLAMF1^{-/-}$), SLAMF2-deficient ($SLAMF2^{-/-}$), SLAMF3-deficient ($SLAMF3^{-/-}$), SLAMF5-deficient ($SLAMF5^{-/-}$), SLAMF6-deficient ($SLAMF6^{-/-}$), SLAMF7-deficient ($SLAMF7^{-/-}$), SLAMF3 and SLAMF5-double deficient ($SLAMF3/5^{-/-}$), IL-13-deficient ($IL-13^{-/-}$), SAP and EAT-2-double deficient ($SAP^{-/-}EAT-2^{-/-}$) mice were generated using CRISPR-Cas9-based genome editing in our lab[26,54]. *CD11c-DTR* mice[55] were a gift from Zhihua Liu (Tsinghua University, Beijing, China). *OT-II* transgenic (#004194), *Rag1^{-/-}* (#002216), *CD45.1* (#002014), and C57BL/6 (#000664) mice were obtained from the Jackson Laboratory (Bar Harbor, Maine, USA). Both male and female mice around 8 weeks old on a C57BL/6 background were used in all experiments. Mice were

**Fig. 5 | The bystander T cells inhibit ILC2 activation in the MLN through SFRs. a** The tSNE analysis was performed on SFR-positive cells gated by flow cytometry from the MLN of WT mice at day 6 following three consecutive days of papain treatment (top and middle panels). The bottom panel displays the average proportion of SFR-positive cells. $1 \times 10^4$ WT (**b**) or SFR-deficient (**c**) ILC2s were stimulated with or without $1 \times 10^5$ WT or SFR-deficient T- or B-cells in the presence of IL-7 plus IL-33 for 48 h. Following a 3-h restimulation with PMA plus ionomycin, the expression of IL-5 and IL-13 by ILC2s was analyzed using intracellular staining. $n = 5$, 4, 4, 4, 3 (**b**) and $n = 6$, 6, 4, 5, 3 (**c**) mice in groups shown from left to right. **d, e** After a 3-h PMA plus ionomycin restimulation on day 6, the intracellular staining was performed to detect the expression of IL-13 in MLN $CD45.1^-CD45.2^+DsRed^-$ ILC2s from the mice (as illustrated in Supplementary Fig. 5g) following papain (**d**) or IL-33 (**e**) treatment (i.n., day 0, 1 and 2). $n = 4$, 5 mice (left to right groups, **d**); $n = 3$ mice

per group (**e**). **f** Representative immunofluorescence staining (left) and quantitation (right) of MLN ILC2s in mice on day 6 after papain treatment (i.n., day 0, 1 and 2). Green: CD3, representing T zone; Red: B220, representing B zone; Sky blue sphere: KLRG1-positive and CD3-negative, representing ILC2s. $n(WT) = 74$ and $n(SFR^{-/-}) = 78$ field per group. **g** After a 3-hour PMA plus ionomycin restimulation on day 6, the intracellular staining was performed to detect the expression of IL-5 and IL-13 in MLN ILC2s from mice following IL-33 treatment (i.n., day 0, 1 and 2). $n = 11$, 9, 7, 11, 7 mice (left to right groups). The data represent at least two independent experiments with similar results. All data are represented as means ± SEM, and statistical analysis was conducted using one-way ANOVA (**f**, **g**) or two-tailed Student's $t$ test (**b**–**e**). ns, not significant. Please refer to Supplementary Fig. 5 for further details.

---

euthanized by carbon dioxide inhalation. All experimental/control animals were housed under the same conditions. All mice were maintained in specific pathogen-free animal facilities at Tsinghua University. All experiments involving animals were conducted with the explicit permission granted by the Animal Ethics Committee of Tsinghua University, in accordance with their official guidelines and regulations.

### Reagents
Antibodies against CD3e (145-2C11, F 1:500), CD3 (17A2, IF 1:200), CD19 (eBio1D3, F 1:500), NK1.1 (PK136, F 1:500), CD11b (M1/70, F 1:500), MHC-II (M5/114.15.2, F 1:500), ST2 (RMST2-2, F 1:500), Siglec F (1RNM44N, F 1:500), SLAMF2 (HM48-1, F 1:500), SLAMF4 (eBio244F4, F 1:500), CD45.1 (A20, F 1:500), CD45.2 (104, F 1:500), CD4 (GK1.5, F 1:500), CD11c (N418, F 1:500), CD8 (53-6.7, F 1:500), B220 (RA3-6B2, IF 1:200), Gr1 (RB6-8C5, F 1:500), KLRG1 (2F1, F 1:500, IF 1:200), CD90.2 (53-2.1, F 1:500), CD90.1 (HIS51, F 1:500), IL-4 (11B11, F 1:250), IL-5 (TRFK5, F 1:250), IL-13 (eBio13A, F 1:250), GATA-3 (TWAJ, F 1:100), CD69 (H1.2F3, F 1:500), CD44 (IM7, F 1:500), Syrian Hamster IgG (H + L) (#A21451, IF 1:200), Rat IgG (H + L) (#A11006, IF 1:200), Rabbit IgG (H + L) (#A11011, IF 1:200), Rabbit IgG (H + L) (#A21244, F 1:500), Mouse IgG (H + L) (#A16067, F 1:500), Streptavidin (#S21388, IF 1:200, F 1:500) were from Thermo Fisher Scientific. Antibodies against SLAMF1 (TC15-12F12.2, F 1:500), SLAMF3 (ly9ab3, F 1:500, IF 1:200), SLAMF5 (mCD84.7, F 1:500, IF 1:200), SLAMF6 (330-AJ, F 1:500), SLAMF7 (4G2, F 1:500) and recombinant mouse IL-33, IL-7 were obtained from Biolegend. Antibodies against SAP (1A9, F 1:200) were obtained from BD Biosciences. Lineage markers contain CD3e, CD19, CD11b, NK1.1, and Gr1. IκBα (L35A5, F 1:200), P-IκBα (Ser32/36) (5A5, WB 1:1000, F 1:200), P-NF-κB p65 Ser536 (93H1, IF 1:200, WB 1:1000), P-SHP1 Tyr564 (D11G5, F 1:200), P-SHP2 Tyr542 (E8D6V, F 1:200), P-SHIP1 Tyr1020 (F 1:200, WB 1:1000), SHP-1 (E1U6R, F 1:200), SHP-2 (D50F2, F 1:200), SHIP1 (E8M5D, F 1:200, WB 1:1000) and GAPDH (D16H11, WB 1:1000) were from cell signaling technology. Papain was from Sigma-Aldrich. PE labeled tetramer of I-A(b) mouse 2W1S EAWGALANWAVDSA and APC labeled tetramer of I-A(b) chicken ova 325-335 QAVHAAHAEIN were provided by the NIH and used at 1:250 dilution. collagenase D and DNase I were from Roche. The peptide OVA323 (ISQAVHAAHAEI-NEAGR) and peptide 2W1S (EAWGALANWAVDSA) were from Scilight Biotechnology LCC. Clophosome (clodronateloaded liposomes) were purchased from FormuMax. SR3335 and FTY720 were from MedChemExpress.

### Papain or IL-33 or A. alternata treatment
Mice were anesthetized with isoflurane and then received intranasal administration of papain (10 μg in 40 μL PBS) or IL-33 (500 ng in 40 μL PBS) or A. alternata (30 μg in 40 μL PBS) on the indicated days. BALF, lungs and MLN were collected and analyzed on the indicated days.

### Induction of tetramer-traceable antigen-specific CD4$^+$ T cells
Tetramer-traceable antigen-specific CD4$^+$ T cells were generated by the intranasal administration of the peptide 2W1S (50 μg) or peptide

OVA323 (50 μg) together with papain (10 μg) or IL-33 (500 ng) to mice on day 0, 1 and 2[19,56]. On day 9, 2W1S:I-Ab MHCII tetramer or ova325-335:I-Ab MHCII tetramer were used to stain antigen-specific CD4$^+$ T cells from the lungs.

### Flow cytometry
Cell surface markers were stained with antibodies) in FACS buffer ($1 \times$ PBS + 2% FBS + 0.02% NaN3). For measurement of intracellular cytokine expression, cells were isolated ex vivo and stimulated with 50 ng/mL PMA and 1 mM ionomycin in the presence of monensin and Brefeldin A (eBioscience) for 3 h. For detection of phosphorylated signaling proteins, ILC2s were fixed with Phosflow Lyse/Fix buffer, followed by permeabilization with Phosflow Perm buffer III (BD), and stained with the indicated antibodies (Cell Signaling Technology). Flow cytometry data were collected using the LSR Fortessa flow cytometer (BD Biosciences) and analyzed with FlowJo software. The net mean fluorescence intensity (MFI) was calculated.

### Histology
For immunofluorescence staining, lungs or MLN were fixed in 4% paraformaldehyde, embedded in O.C.T. compound and sliced into 15 μm frozen sections. The slides were fixed with 1% PFA and then blocked/permeabilized by PBS with 5% BSA and 0.5% triton X 100 (Sigma-Aldrich). Following this, the slides were incubated for 1 h at room temperature with specific primary antibodies in permeabilization buffer (PBS with 5% BSA and 0.2% triton X 100), washed twice, and incubated overnight at 4 °C with the fluorescently labeled secondary antibody in permeabilization buffer. Lastly, the slides were stained with DAPI. Confocal microscopy was performed with Nikon AX. Images were processed with the NIS-Elements software. For hematoxylin and eosin (H&E) staining of the lung tissue, lungs were fixed in 4% phosphate-buffered formalin and embedded in paraffin for sectioning. Sections (5 mm) were used for H&E staining. Images were acquired using Axio Scan. Z1 Zeiss. The images were analyzed with Zeiss software. Anti-KLRG1 (Thermo, #16-5893-82), anti-CD3 (Thermo, #16-0032-82), anti-B220 (Thermo, #12-0452-82), anti-Syrian Hamster IgG (H + L) (Thermo, #A21451), anti-Rat IgG (H + L) (Thermo, #A11006), anti-Rabbit IgG (H + L) (Thermo, #A11011), Streptavidin-PE (Thermo, #S21388), anti-SLAMF3 (Biolegend, #122903), anti-SLAMF5 (Biolegend, #122803), anti-P-NF-κB p65 (CST, #3033) were used for immunofluorescence staining.

### Tissue preparation and immune cell isolation
Cell collection from the BALF and lungs was prepared as previously described[57]. Briefly, for cell collection from BALF, the lung tissues were lavaged with 800 μL of cold PBS three times, and the supernatant was collected. To isolate cells from the lungs, the lung tissues were first minced and digested using 2 mg/mL collagenase D (Roche) and 20 μg/mL DNase I (Roche) in RPMI-1640 medium with 5% FBS. The digestion was carried out on a shaker at 180 rpm for 45 min at 37 °C. Subsequently, the digested tissue was mashed through 70-mm cell

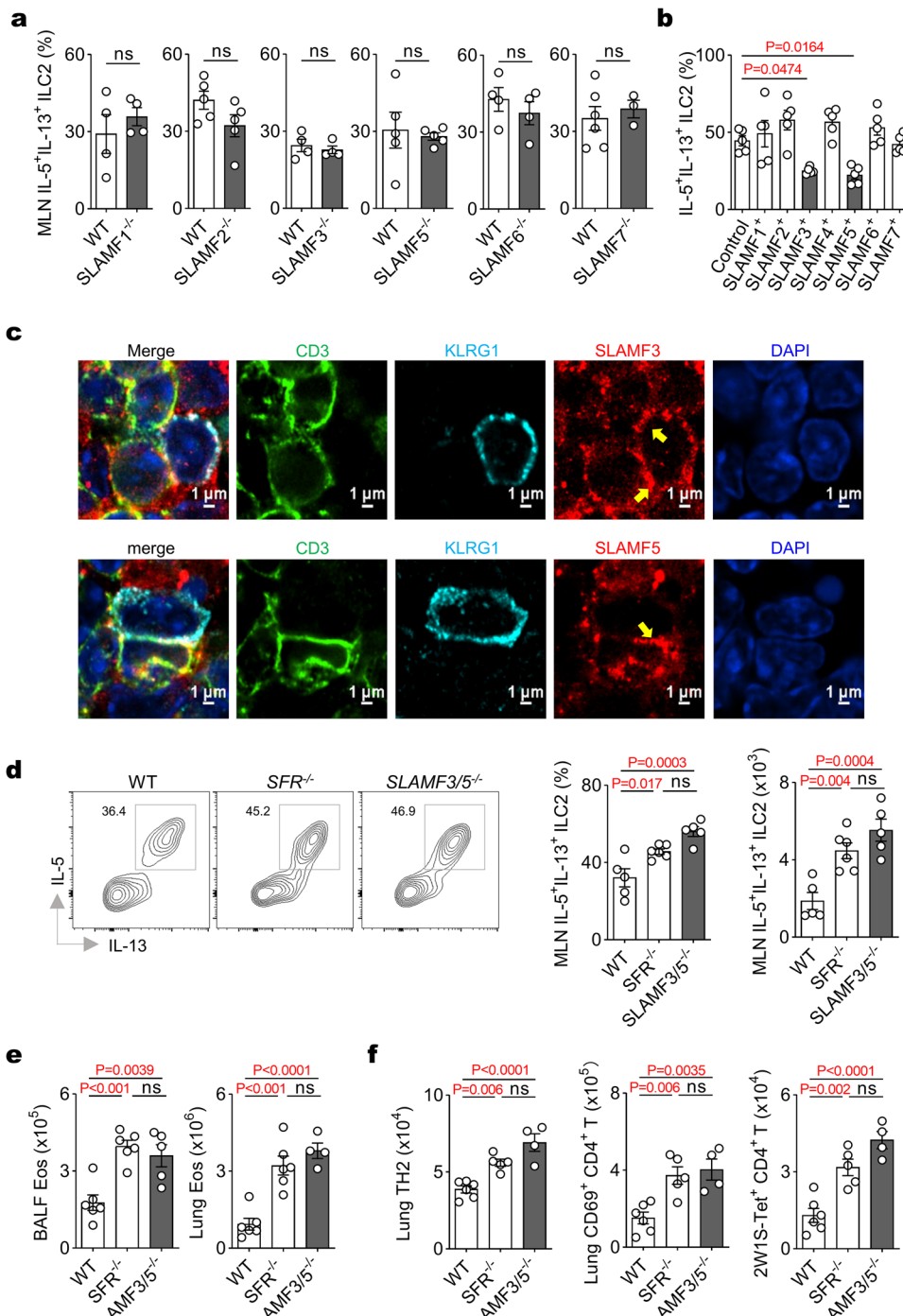

**Fig. 6 | Two SFR members, namely SLAMF3 and SLAMF5, suppresses ILC2 activation in MLN. a** After 3 h of PMA plus ionomycin restimulation on day 6, the intracellular staining was performed to detect the expression of IL-5 and IL-13 in MLN ILC2s from mice following IL-33 treatment (i.n., day 0, 1 and 2). $n$ = 4, 4, 5, 5, 4, 4, 5, 5, 4, 4, 6, 3 mice (left to right groups). **b** WT ILC2s were stimulated with OP9-DL1 cells ectopically expressing individual SFR member in the presence of IL-7 plus IL-33 for 72 h. Following a 3-h restimulation with PMA plus ionomycin, the expression of IL-5 and IL-13 by ILC2s was analyzed using intracellular staining. $n$ = 5 per group. **c** Confocal microscopic analysis of the polarization of SLAMF3 or SLAMF5 (red) on the contact surface (yellow arrow) between ILC2s (KLRG1, sky blue) and their bystander T cells (CD3, green) within the MLN from mice on day 6 following IL-33 treatment (i.n., day 0, 1 and 2). **d** Representative

flow cytometric profiles (left) and quantification (right) showing IL-5 and IL-13 expressing in gated MLN ILC2s after 3 h of PMA plus ionomycin restimulation on day 6 following IL-33 treatment (i.n., day 0, 1 and 2) in the indicated mice. $n$ = 5, 6, 5 mice (left to right groups). The indicated mice were treated with IL-33 plus 2W1S peptide (i.n., day 0, 1 and 2), and the numbers of eosinophils (**e**), TH2 cells, CD69+ CD4+ T cells, and tetramer+ CD4+ T cells (**f**) were analyzed on day 9. $n$ = 6, 6, 5, 6, 6, 4 (**e**) and $n$ = 6, 5, 4, 6, 5, 4, 6, 5, 4 (**f**) mice in groups shown from left to right. The data represent two (**a–c**) or at least three (**d–f**) independent experiments with similar results. All data are represented as means ± SEM, and statistical analysis was conducted using a two-tailed Student's $t$ test (**a**) or one-way ANOVA (**b**, **d–f**). ns not significant. Please refer to Supplementary Fig. 6 for further details.

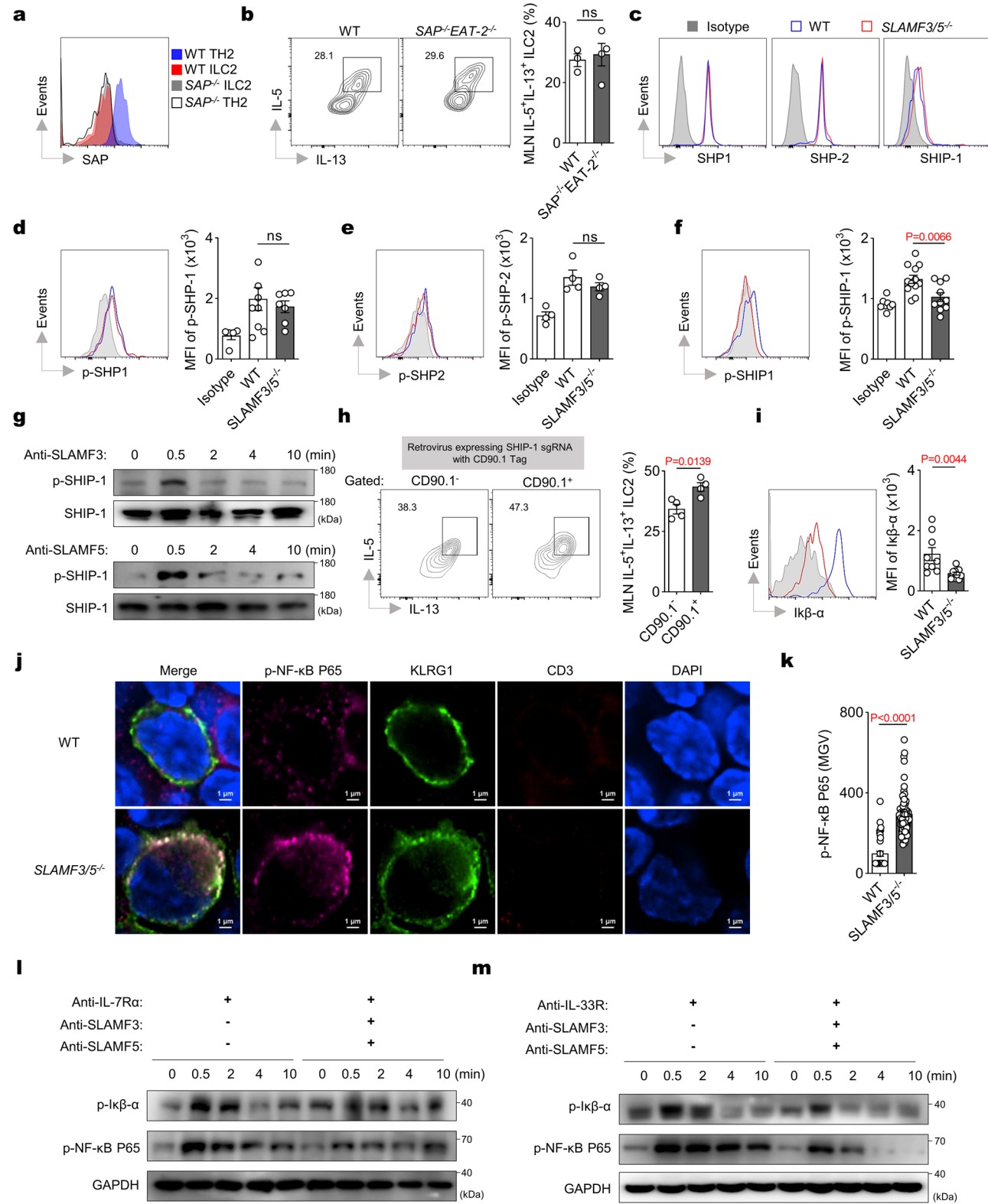

strainers to obtain a single-cell suspension before lysing the red blood cells. For isolation of cells from the MLN, tissues were manually homogenized using 70 μm cell strainers.

**In vitro stimulation of ILC2**

For the western blot analysis of ILC2 signaling, the lung ILC2s were sorted from IL-33-treated *Rag1*^−/− mice and cultured for 5 days in RPMI-1640 complete medium (100 U/mL of penicillin-streptomycin, and 10%

FBS) with 200 U/mL hIL-2. 12 h before stimulation, the ILC2s were washed and cultured in IL-2-null RPMI-1640 complete medium. Subsequently, biotinylated anti-mouse IL-7Rα or IL-33R antibody (10 μg/ml), with or without biotinylated anti-mouse SLAMF3 and/or SLAMF5 antibody (10 μg/ml), were added to the IL-2-expanded ILC2s for 30 min at 4 °C. After washing steps, the ILC2s were incubated with streptavidin (20 μg/ml) for the indicated times at 37 °C. Additionally, IL-2-expanded ILC2s were stimulated with or without IL-7 (15 ng/ml) or

**Fig. 7 | SLAMF3 and SLAMF5 inhibit the downstream NF-κB pathway of IL-7/IL-33 through SHIP-1. a** Detection of SAP expression in the indicated cells on day 9 following IL-33 treatment (i.n., day 0–2). **b** Detection of IL-5 and IL-13 expression in MLN ILC2s on day 6 following IL-33 treatment (i.n., day 0–2). n(WT) = 3 and n(SAP^-/-EAT-2^-/-) = 4 mice. **c** Detection of SHP-1, SHP-2, and SHIP-1 expression in MLN ILC2s on day 6 following IL-33 treatment (i.n., day 0–2). Isotype, fluorescently labeled secondary antibody, representing negative control. Detecting of SHP-1 (**d**), SHP-2 (**e**), and SHIP-1 (**f**) phosphorylation in MLN WT (Blue line) and *SLAMF3/5^-/-* (red line) ILC2s on day 6 following IL-33 treatment (i.n., day 0–2). Gray shade, isotype. n = 5, 9, 7 (**d**) or 3, 3, 3 (**e**) or 8, 12, 10 (**f**) mice (left to right groups). **g** Western blot analysis of the phosphorylated SHIP-1 (P-SHIP-1) in WT ILC2s with antibody-mediated cross-linking of SLAMF3 (top) and SLAMF5 (bottom) at the indicated minute points (min). **h** Detection of IL-5 and IL-13 expression in MLN ILC2s of the CHIME chimera (refer to the methods) on day 6 following IL-33 treatment (i.n., day 0 to 2). CD90.1-

negative: representing sgRNA-unexpressed control; CD90.1-positive: representing SHIP-1 sgRNA-expressed. n = 4 mice per group. **i** Representative overlaid histograms and quantification of Iκβ-α expression in MLN WT (Blue line) and *SLAMF3/5^-/-* (red line) ILC2s on day 6 following IL-33 treatment (i.n., day 0 to 2). Gray shade, representing isotype. n(WT) = 9 and n(*SLAMF3/5^-/-*) = 11 mice. Representative images (**j**) and quantification (**k**) of the phosphorylation of NF-κB in MLN ILC2s from the indicated mice on day 6 following IL-33 treatment (i.n., day 0 to 2). n(WT) = 40 and n(*SLAMF3/5^-/-*) = 70 cells. **l, m** Western blot analysis of NF-κB activation in WT ILC2s stimulated with (+) or without (−) antibodies targeting IL-7Rα (**l**) or IL-33R (**m**), in the presence (+) or absence (−) of antibodies against SLAMF3 or SLAMF5. The data represent at least two independent experiments with similar results. All data are represented as means ± SEM, and statistical analysis was conducted using two-tailed Student's *t* test (**b, h, i, k**) or one-way ANOVA (**d–f**). ns not significant. Please refer to Supplementary Fig. 7 for further details.

IL-33 (15 ng/ml), in the presence or absence of plate-bound anti-SLAMF3 and anti-SLAMF5 antibodies, for 30 min at 37 °C prior to western blot analysis. For in vitro ILC2s and T/B cell co-culture, the indicated number of ILC2s were cultured with or without the specified number of T- or B-cells in RPMI-1640 complete medium with 15 ng/ml IL-7 and 15 ng/ml IL-33 in a 96-well cell culture plate (U bottom) for 48 h. Following a 3-h restimulation with PMA plus ionomycin, the expression of IL-5 and IL-13 by ILC2s was analyzed using intracellular staining. For in vitro ILC2s and OP9-DL1 co-culture, OP9-DL1 cells respectively expressing SLAMF1 to SLAMF7 were initially generated using the lentivirus vector pWPXLd-GFP. Subsequently, flow cytometry-sorted $1 \times 10^4$ lung ILC2s were culture on those OP9-DL1 cells in RPMI-1640 complete medium with 15 ng/ml IL-7 and 15 ng/ml IL-33 in a 96-well cell culture plate (flat bottom) for 72 h. ILC2s were then re-stimulated with PMA plus ionomycin for 3 h, and the expression of IL-5 and IL-13 by ILC2s was analyzed using intracellular staining.

### Adoptive transfer of ILC2
IL-33-treated WT and *SFR^-/-* mice were used to sort lung ILC2s, which were then cultured for 6 or 7 days in RPMI-1640 complete medium (containing 100 U/mL of penicillin-streptomycin, and 10% FBS) supplemented with 200 U/mL hIL-2. Subsequently, $1 \times 10^6$ IL-2-expanded ILC2s from either WT or *SFR^-/-* mice were intravenously transferred into *Nfil3^-/-* recipients. Prior to the transfer, the recipients were depleted of macrophages to prevent macrophage-mediated rejection of SFR-deficient hematopoietic grafts[54]. After the injection, the mice were administrated papain 5 days later as indicated.

### FTY720 treatment
Mice received intranasal administration of papain or IL-33, along with intraperitoneal administration of either DMSO (as a control) or FTY720 (0.5 mg/kg per mouse), for the initial 3-day period. Following this, the mice were administered DMSO or FTY720 intraperitoneally once every 2 days.

### ILC2, CD4+ T cell, DC, NK cell, macrophage depletion
To deplete ILC2s, mice were subjected to daily intraperitoneal injections of 100 μL DMSO (as a control) or the RORα inhibitor SR3335 (100 μg per mouse)[40,41], starting on the 7th day preceding the administration of papain or IL-33. To deplete CD4+ T cells, mice were intraperitoneally injected with 350 μg of the anti-CD4 depleting antibody (clone GK1.5) three times per week prior to the study. To deplete DCs, CD11c-DTR mice were intraperitoneally injected with DTx (100 ng per mouse) every 2 days, starting on the 7th day prior to the 3 consecutive days of papain or IL-33 administration. To deplete NK cells, mice were subjected to intraperitoneal injections of 50 μg of anti-NK1.1 antibody (clone PK136) three times a week, starting on the 7th day before the administration of papain or IL-33. For macrophage depletion, mice received a single intraperitoneal injection of 200 μL of clophosome during the first week, followed by subsequent 100 μL doses every four

days. Papain administration was initiated seven days after the initial clophosome injection.

### BMDC-T cell co-culture
To prepare BMDCs, bone marrow cells were cultured for 7 days in RPMI-1640 complete medium with GM-CSF (10 ng/mL) and IL-4 (10 ng/ml). CD11c^hi cells were sorted and mixed with flow cytometry-sorted naïve OT-II CD4+ T cells at a ratio of 1:10 in the presence of peptide OVA323. The CD4+ T cells were then harvested at day 5 for intracellular staining of IL-4 after 3 h of PMA plus ionomycin restimulation.

### Generation of bone marrow chimera
Mixed bone-marrow (BM) chimera with the indicated genotypes were generated by reconstituting lethally irradiated CD45.1 WT recipients with $6 \times 10^5$ indicated mixed BM cells. After 16 weeks reconstitution, the mice were subjected to be analyzed before the indicated treatment.

### CHIME chimera setup
The flow cytometry-sorted LSK (lineage^-Sca-1+Kit+) cells derived from bone marrow of cas9 transgenic mice were subjected to spin-transduction with MCMV retrovirus-containing supernatants for a duration of 120 min, under conditions of 1500 g force and at a temperature of 32 °C. The MCMV retrovirus used in the transduction carried the sgRNA (AGGGCTCAGAATCTACCAAC), which specifically targets the SHIP-1 gene, along with the CD90.1 tag. Subsequently, these transduced LSK cells were intravenously introduced into CD45.1 WT recipients that had been lethally irradiated (10 Gy), following the protocol described in refs. 43, 44. After 16 weeks, the mice were euthanized for further analysis.

### Gene-expression analysis
The scRNA-seq dataset of lung ILC2s with different treatments in Supplementary Figs. 1a, b and 7a, b was obtained from the Gene Expression Omnibus (GEO; GSE102299 and GSE131996)[58,59] and then processed using the R package Seurat[60]. Only the 10x scRNA-seq from WT mice (GSM2733478 to GSM2733491 and GSM4192253 to GSM4192257) were used.

### Statistical analyses
Prism 8 software was used for performing one-way ANOVA, two-way ANOVA, unpaired two-tailed Student's *t*-tests, and linear regression analyses. For comparison of one group, *P* values were calculated using two-tailed Student's *t* tests. For comparison of more than one group, *P* values were calculated using one-way or two-way ANOVA. All data are shown as means ± SEM. Differences with a *p*-value below 0.05 were considered statistically significant.

### Reporting summary
Further information on research design is available in the Nature Portfolio Reporting Summary linked to this article.

## Data availability

The previously published scRNA-seq data of GSE102299 (specifically GSM2733478 to GSM2733491, https://www.ncbi.nlm.nih.gov/geo/query/acc.cgi?acc=GSE102299) and GSE131996 (specifically GSM4192253 to GSM4192257, https://www.ncbi.nlm.nih.gov/geo/query/acc.cgi?acc=GSE131996) from the Gene Expression Omnibus database were used in this paper. The remaining data are available within the Article or Source Data file. Source data are provided with this paper.

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

## Acknowledgements

The research presented in this publication was financially supported by the Natural Science Foundation of China (to Z.D., grant number, 32330034; S.C., grant number, 82371734, and to D.L., grant number, 82103327); National Key Research & Developmental Program of China (grant number 2022YFF0710602). Anhui Provincial Department of Education (Grant No.2023AH010085 to Z.D., 2022AH030114 to S.C.), and Tsinghua University Initiative Scientific Research Program of Precision Medicine (2023ZLA002).

## Author contributions

Y.W. conducted the majority of experiments, performed data analysis, and drafted the manuscript. Y.Q. and J.H. conceived the project and discussed the data. Z.D. conceptualized the study, conducted data analysis, and co-wrote the manuscript with assistance from Y.W. and S.C.

## Competing interests

The authors declare no competing interests.
