## [Peer Review File · Nature Communications]

SLAM-family receptors promote resolution of ILC2-mediated inflammationEditorial Note: Parts of this Peer Review File have been redacted as indicated to maintain the confidentiality of unpublished data.

REVIEWER COMMENTS

Reviewer #1 (Remarks to the Author):

in this tour-de-force set of experiments the authors show in a convincing manner that ILC2s are regulated negatively by SLAMF3/5 expressed by T cells in the MLNs. They have used a large number of KO and cKO mice to support their conclusions. The presentation is clear and in a proper laconic style.

I have no comments for the authors except for entering a sentence or two on the possible translational value of their findings.

Reviewer #2 (Remarks to the Author):

This manuscript provides evidence for a function of SLAM-family receptor in ILC2. The authors further propose that SLAM-family receptors regulate the crosstalk between ILC2-DC-T cells after papain treatment of IL-33 administration. The authors generated an impressive amount of knockout mice, BM chimeras, and data. However, the results are complicated to evaluate due to the mouse model used.

Most experiments were conducted with full knockouts of SFR or associated molecules. Therefore, these experiments do not exclude a direct effect of the SFR knockout on T cells (or other cell types), for which a function was already described. In fact, some of the data shown from SFR^{-/-} Rag1^{-/-} and FTY720 experiments argue for a function of SFR in T cells. I understand that the interpretation in the manuscript is that this should reflect ILC2 -T cell regulation, but this cannot be concluded from these experiments. Alternatively, SFR could directly regulate T cells or DC without the involvement of ILC2.

In general, it is difficult to assess from the data shown if SFR are highly expressed in ILC2s compared to other immune cells.

SFR^{-/-} mice were crossed to Nfil3^{-/-} mice to address the question of SFR function in other immune cells. However, Nfil3 is broadly expressed in the immune system, including ILC, T cells and DC (PMID: 21474667,31311918). Therefore, this mouse model does not exclude effects on T cells and DC.

As a complementary approach, RORa inhibitors were used, but RORa is functionally important in many immune cells. This involves all ILC subsets (PMID: 31676672, 34556887, 31128961) and T cells/Th2 cells (PMID: 33165810, 34003838) and RORa inhibitors are not suitable to demonstrate a specific function in ILC2.

To sum up, I am not confident to tell in which cell type SLAM-family receptors play a role based on the data provided.

Reviewer #3 (Remarks to the Author):

The study presented by Wang and colleagues is a comprehensive piece of work addressing the role of SLAM-family receptors in ILC2s during type-2 lung inflammation. Whilst previous work by the authors and other groups identified a role for SFRs in NK cell biology, no data were available regarding other types of innate lymphoid cells, such as ILC2s. The authors' careful kinetics analysis of the response of SFR-deficient mice to inhalation of type-2 stimuli allowed them to suggest the involvement of the SFR members SLAMF3 and SLAMF5 as key signalling modules between CD4 T cells and ILC2s, thereby limiting ILC2 activation in the mLN and subsequent lung eosinophilia. The authors' working model was notably supported by sophisticated mouse genetic models that

allowed them to pinpoint the cellular and molecular components underpinning their initial observations. I found that the analysis presented was thorough although limited to 2 basic models of type-2 lung inflammation (papain and IL-33). Thus, the larger significance of the work is not obvious given the choice of the models (papain and IL-33); yet given the relative paucity of data concerning this family of self-ligands/receptors, this study provides essential fundamental research of quality in this topic. The main limitation of the work is discussed by the authors and concerns the lack of a conditional deletion of the SFRs in ILC2s, which will be the focus of future studies.

Specific questions:

1/ Related to the choice of model. The author's observations very specifically showed the effect of SFR deficiency at day 6 in the mLN and day 9 in the lung following papain or IL-33 exposure. By day 9 the effect is normalised in the mLN. For the lung, we don't know if the eosinophilia persists much longer than day 9. Did the authors think of trying a sensitisation/challenge model for papain instead? For instance, exposure on d0, d1 and d14, then analysis on d20? Or a more physiologically relevant allergen model such as *A. Alternata* or House Dust Mite?

2/ Related to SLAM antibody stainings (Figure 1 and Figure 5/S5). None of the conditions presented show positive staining for SLAMF4. Is this antibody showing positive staining in other cell types (not presented here) i.e. can we be sure that this antibody works/is specific? SLAMF7 staining is also not showing a positive staining, except in Figure S5J. Strangely there we see staining in B cells WT, lost in SFR KO; however, this is not what was shown above for similar conditions in Figure S5A, where we can't see any staining in WT B cells. Could the authors clarify this discrepancy – can the nature of the stimuli (papain versus IL-33) really be the cause of this difference ?

3/ Related to the co-cultures presented in Figure 5B (and C). As far as I understand the material and methods, the "ILC2-only" condition is with 1×10^4 cells, and the co-culture is an addition of 1×10^5 cells. I am worried that this is not a fair comparison, given the influence of media/growth factor consumption on cell metabolism and cytokine production. Could the authors repeat these co-cultures and provide data adding a condition comprising ILC2s co-cultured with "themselves" to reach the same total amount of cells, or repeat the same ratio as used in the Figure but with 5 to 10 times less cells in the wells (providing it still allows them to quantify cytokine production by ILC2s).

4/ Regarding the IF data (Figure 5F, 6C, 7J). I am not convinced by the identification of ILC2s solely on CD3-KLRG1+ cells. As far as I am aware, NK cells for instance could also be found in this definition. Could the authors use a Gata3 staining to ascertain their data where possible (by multiplex IF for instance)? Related to this, the authors should mention which antibodies were used for flow and which were used for IF in their material and methods (or add the IF antibodies in the relevant section).

5/ Regarding the Western blotting (Figure 7G, 7L, 7M). Could the authors provide quantifications of the bands to support their conclusions? If possible, the authors should also repeat the pSHIP-1 WB with the anti-SLAMF5 (Figure 7G), as it is particularly dirty and does not allow for a proper visual examination of the bands. Whilst I agree that SLAMF3 crosslinking induces SHIP1 phosphorylation, it is more difficult to conclude at this point for SLAMF5. Could the authors also include in the legend if the ILC2s were also stimulated with the anti-IL7ra and anti-IL33R in this case as in Figure 7L/M ? Related to that, I am not sure to understand the rationale for using antibodies for ILC2 stimulation. Could the authors explain why they did not simply add recombinant IL-33 and IL-7 to their cultures instead?

Minor comment:

Figure 7I: the label of the axis MFI lacks the $(\times 10^3)$

Point-to-point Response to Reviewers

Reviewer #1 (Remarks to the Author):

in this tour-de-force set of experiments the authors show in a convincing manner that ILC2s are regulated negatively by SLAMF3/5 expressed by T cells in the MLNs. They have used a large number of KO and cKO mice to support their conclusions. The presentation is clear and in a proper laconic style.

I have no comments for the authors except for entering a sentence or two on the possible translational value of their findings.

Response: Please refer to the last sentence in the Discussion section of the revised manuscript.

Reviewer #2 (Remarks to the Author):

This manuscript provides evidence for a function of SLAM-family receptor in ILC2. The authors further propose that SLAM-family receptors regulate the crosstalk between ILC2-DC-T cells after papain treatment of IL-33 administration. The authors generated an impressive amount of knockout mice, BM chimeras, and data. However, the results are complicated to evaluate due to the mouse model used.

Most experiments were conducted with full knockouts of SFR or associated molecules. Therefore, these experiments do not exclude a direct effect of the SFR knockout on T cells (or other cell types), for which a function was already described. In fact, some of the data shown from SFR^{-/-} Rag1^{-/-} and FTY720 experiments argue for a function of SFR in T cells. I understand that the interpretation in the manuscript is that this should reflect ILC2-T cell regulation, but this cannot be concluded from these experiments. Alternatively, SFR could directly regulate T cells or DC without the involvement of ILC2.

In general, it is difficult to assess from the data shown if SFR are highly expressed in ILC2s compared to other immune cells.

Response: We express our appreciation to the reviewer for their valuable comments, and we agree with the concerns raised.

It is indeed important to acknowledge the potential impact of SFRs on T cells, as our unpublished data indicate an inherent promotion of TH2 differentiation by SFRs (Please refer to the figure below). Interestingly, in the context of our experimental models, we observe an inhibitory effect of SFR on TH2 responses (Please see Fig. 3E and S3E). This inconsistency may suggest that the intrinsic role of SFR in promoting TH2 responses and

subsequently increasing eosinophilia might be limited, at least within the papain or IL-33-induced models where ILC2s are predominant. Furthermore, our in vitro experiments with BMDC-induced TH2 cells and in vivo bone marrow chimeric experiments using DC lacking SFR also provide evidence against the direct regulatory role of SFR on DC in increasing eosinophilia (Please see Fig. S4G and S4J).

We acknowledge the use of T cell- or ILC2-specific knockout mice. However, we faced challenges in efficiently achieving deletion due to the large size of the floxed SFR loci (approximately 400 kb) when attempting to generate SFR^{fl/fl}/IL5-Cre and SFR^{fl/fl}/CD11c-Cre mice. Despite these challenges, our ongoing efforts are directed towards generating SLAMF3^{fl/fl}/SLAMF5^{fl/fl} mice for future investigations. Reviewer #3 also acknowledges the difficulties we encountered.

Nevertheless, as an alternative approach, although it is difficult to completely exclude the intrinsic impact of SFR on T cells and DCs, we present new evidence through adoptive transfer experiments to elucidate the influence of ILC2-specific SFR deficiency on TH2 responses and DCs. We observed that the transfer of SFR-deficient ILC2s into Nfil3^{-/-} recipients led to increased TH2 responses and MLN DC counts compared to the transfer of WT ILC2s after papain treatment (Please refer to Fig. S2E, S3O, and S4K). This supports the roles of SFRs on ILC2s in shaping the behaviors of TH2 cells and DCs.

[REDACTED]

SFR^{-/-} mice were crossed to Nfil3^{-/-} mice to address the question of SFR function in other immune cells. However, Nfil3 is broadly expressed in the immune system, including ILC, T cells and DC (PMID: 21474667,31311918). Therefore, this mouse model does not exclude effects on T cells and DC. As a complementary approach, RORa inhibitors were used, but RORa is functionally important in many immune cells. This involves all ILC subsets (PMID: 31676672, 34556887, 31128961) and T cells/Th2 cells (PMID: 33165810, 34003838) and RORa inhibitors are not suitable to demonstrate a specific function in ILC2. To sum up, I am not confident to tell in which cell type SLAM-family receptors play a role based on the data provided.

Response: As mentioned earlier, we present new experimental evidence through adoptive transfer to confirm the specific roles of SFRs in ILC2s in shaping the behaviors of TH2 cells and DCs (Please refer to Fig. S2E, S3O, and S4K).

Reviewer #3 (Remarks to the Author):

The study presented by Wang and colleagues is a comprehensive piece of work addressing the role of SLAM-family receptors in ILC2s during type-2 lung inflammation. Whilst previous work by the authors and other groups identified a role for SFRs in NK cell biology, no data were available regarding other types of innate lymphoid cells, such as ILC2s. The authors' careful kinetics analysis of the response of SFR-deficient mice to inhalation of type-2 stimuli allowed them to suggest the involvement of the SFR members SLAMF3 and SLAMF5 as key signalling modules between CD4 T cells and ILC2s, thereby limiting ILC2 activation in the mLN and subsequent lung eosinophilia. The authors' working model was notably supported by sophisticated mouse genetic models that allowed them to pinpoint the cellular and molecular components underpinning their initial observations. I found that the analysis presented was thorough although limited to 2 basic models of type-2 lung inflammation (papain and IL-33). Thus, the larger significance of the work is not obvious given the choice of the models (papain and IL-33); yet given the relative paucity of data concerning this family of self-ligands/receptors, this study provides essential fundamental research of quality in this topic. The main limitation of the work is discussed by the authors and concerns the lack of a conditional deletion of the SFRs in ILC2s, which will be the focus of future studies.

Specific questions:

1/ Related to the choice of model. The author's observations very specifically showed the effect of SFR deficiency at day 6 in the mLN and day 9 in the lung following papain or IL-33 exposure. By day 9 the effect is normalised in the mLN. For the lung, we don't know if the eosinophilia persists much longer than day 9. Did the authors think of trying a sensitisation/challenge model for papain instead? For instance, exposure on d0, d1 and d14, then analysis on d20? Or a more physiologically relevant allergen model such as *A. Alternata* or House Dust Mite?

Response: We would like to express our gratitude to the reviewer for providing these valuable suggestions. Please find below the corresponding data:

1. In accordance with the recommendation, we performed an *A. Alternata* treatment (i.n. on day 0, 1, and 2) to establish a biologically relevant allergen model. Similar trends were observed as in the case of papain or IL-33 treatment (Please refer to Fig. 1I, S2B, S3F, S3I, and S4E).
2. The sensitization/challenge model induced by papain exhibited increased eosinophilia and a Th2 response in SFR-deficient mice (Please see Fig. S1H and S3J).
3. Similarly, the sensitization/challenge model induced by *A. Alternata* also showed increased eosinophilia and a Th2 response in SFR-deficient mice (Please see Fig. S1I and S3K).

2/ Related to SLAM antibody stainings (Figure 1 and Figure 5/S5). None of the conditions presented show positive staining for SLAMF4. Is this antibody showing positive staining in other cell types (not presented here) i.e. can we be sure that this antibody works/is specific? SLAMF7 staining is also not showing a positive staining, except in Figure S5J. Strangely there we see

staining in B cells WT, lost in SFR KO; however, this is not what was shown above for similar conditions in Figure S5A, where we can't see any staining in WT B cells. Could the authors clarify this discrepancy – can the nature of the stimuli (papain versus IL-33) really be the cause of this difference?

Response:

1. NK cells are SLAMF4-positive cells, and the staining of SLAMF4 on NK cells confirms the efficacy of the anti-SLAMF4 antibody (Please refer to the figure below).

2. We deeply regret the inaccuracies that arose due to the unavailability of a reliable anti-SLAMF7 antibody during the initial stages of this investigation, leading to suboptimal staining for SLAMF7. To address this issue, we meticulously repeated the experiments using a recently commercial anti-SLAMF7 antibody that exhibited superior effectiveness. As a result, we have replaced the corresponding figures with the new ones (Please see Fig. 1A, 1B, 5A, and S5A).

3/ Related to the co-cultures presented in Figure 5B (and C). As far as I understand the material and methods, the "ILC2-only" condition is with 1×10^4 cells, and the co-culture is an addition of 1×10^5 cells. I am worried that this is not a fair comparison, given the influence of media/growth factor consumption on cell metabolism and cytokine production. Could the authors repeat these co-cultures and provide data adding a condition comprising ILC2s co-cultured with "themselves" to reach the same total amount of cells, or repeat the same ratio as used in the Figure but with 5 to 10 times less cells in the wells (providing it still allows them to quantify cytokine production by ILC2s).

Response: We appreciate this suggestion. Three different approaches were employed.

1. The first approach involved repeating the co-cultures with the same total number of cells (Please refer to Fig. S5D).
2. The second approach involved repeating the same cell ratios but with 10 times fewer cells in each well (Please refer to Figure S5E). The observed phenomena were similar to those observed in Fig. 5B and 5C.
3. Additionally, to further eliminate potential confounding effects of the cultural milieu, an equal mixture of WT and SFR-deficient ILC2s was co-cultured with either T or B cells. As anticipated, the cytokine production of SFR-null ILC2s exhibited a significant increase compared to that of WT ILC2s in the presence of co-cultured WT T or B cells (Please refer to Fig. S5F).

4/ Regarding the IF data (Figure 5F, 6C, 7J). I am not convinced by the identification of ILC2s solely on CD3-KLRG1+ cells. As far as I am aware, NK cells for instance could also be found in this definition. Could the authors use a Gata3 staining to ascertain their data where possible (by multiplex IF for instance)? Related to this, the authors should mention which antibodies were used for flow and which were used for IF in their material and methods (or add the IF antibodies in the relevant section).

Response: We deeply appreciate the thoughtful consideration of the reviewer. Despite our persistent efforts, we encountered challenges in effectively using GATA-3 for IF staining on ILC2s. It remains uncertain whether this limitation has resulted in a scarcity of studies specifically utilizing GATA-3 for IF staining on ILC2s, as many studies instead employ CD3-KLRG1+ for ILC2 identification (PMID: 28869965, 28869974, 31353223, 33674322, et al.). As an alternative approach to eliminate the interference of NK cells, we employed an anti-NK1.1 antibody for NK cell depletion and observed that approximately 90% of CD3-KLRG1+ cells in MLN were ILC2s (Please refer to Fig. S5K). Consistent with Figures 5F, 6C, and 7J, a similar phenomenon was still evident after NK cell depletion (Please refer to Figs. S5L, S6D, and S7D).

The detailed information regarding the IF antibodies has been added (Please refer to the Histology section within the Materials and Methods).

5/ Regarding the Western blotting (Figure 7G, 7L, 7M). Could the authors provide quantifications of the bands to support their conclusions? If possible, the authors should also repeat the pSHIP-1 WB with the anti-SLAMF5 (Figure 7G), as it is particularly dirty and does not allow for a proper visual examination of the bands. Whilst I agree that SLAMF3 crosslinking induces SHIP1 phosphorylation, it is more difficult to conclude at this point for SLAMF5. Could the authors also include in the legend if the ILC2s were also stimulated with the anti-IL7ra and anti-IL33R in this case as in Figure 7L/M ? Related to that, I am not sure to understand the rationale for using antibodies for ILC2 stimulation. Could the authors explain why they did not simply add recombinant IL-33 and IL-7 to their cultures instead?

Response:

1. The quantifications of the bands in Figure 7G, 7L, 7M are provided (Please refer to Fig. S7C, S7E, and S7F).
2. The pSHIP-1 WB data for anti-SLAMF5 were replaced (Please see Fig. 7G).
3. The induction of IL-7 and IL-33 signaling through anti-IL7ra and anti-IL33R crosslinking (Fig. 7L and 7M) is similar to the commonly employed anti-CD3 crosslinking strategy in TCR signaling investigations. In line with the recommendation, alternative stimuli using the exogenous presence of recombinant IL-7 or IL-33 have also revealed the suppressive effects exerted by SLAMF3 and SLAMF5 on the IL-7 and IL-33 signaling pathways (Please see Fig. S7I and S7J).

Minor comment:

Figure 7I: the label of the axis MFI lacks the (x10E3)

Response: Done.

REVIEWERS' COMMENTS

Reviewer #1 (Remarks to the Author):

I do not have any comments.

Reviewer #2 (Remarks to the Author):

The authors have acknowledged some of the limitations with respect to conditional gene targeting in their study. I would suggest discussing these limitations as the authors have already started and tone down some of the conclusions regarding an ILC2 -DC - Th2 axis since some of this is based on indirect evidence or adoptive cell transcripts, which are difficult to interpret. If this is fully disclosed, I have no objections to proceed with publication.

Reviewer #3 (Remarks to the Author):

All my concerns were thoroughly addressed by the authors; I don't have any further comments. I support publication of this manuscript.

Point-to-point Response to Reviewers

Reviewer #1 (Remarks to the Author):

I do not have any comments.

Response: We thank the reviewer for their supportive comments on our work.

Reviewer #2 (Remarks to the Author):

The authors have acknowledged some of the limitations with respect to conditional gene targeting in their study. I would suggest discussing these limitations as the authors have already started and tone down some of the conclusions regarding an ILC2 -DC - Th2 axis since some of this is based on indirect evidence or adoptive cell transcripts, which are difficult to interpret. If this is fully disclosed, I have no objections to proceed with publication.

Response: We would like to express our gratitude to the reviewer for providing these valuable suggestions. Please find below the corresponding changes:

1. The mentioned limitations were discussed in the Discussion section (please refer to the fifth paragraph of Discussion section).
2. The conclusions regarding an ILC2 -DC - Th2 axis were toned down thoroughly (please refer to the section of Abstract, Introduction (last paragraph), Results and Discussion).

Reviewer #3 (Remarks to the Author):

All my concerns were thoroughly addressed by the authors; I don't have any further comments. I support publication of this manuscript.

Response: We thank the reviewer for their supportive comments on our work.